# Revealing the Structure of Deep Neural Networks via Convex Duality

## Abstract

We study regularized deep neural networks (DNNs) and introduce a convex analytic framework to characterize the structure of the hidden layers. We show that a set of optimal hidden layer weights for a norm regularized DNN training problem can be explicitly found as the extreme points of a convex set. For the special case of deep linear networks with $K$ outputs, we prove that each optimal weight matrix is rank-$K$ and aligns with the previous layers via duality. More importantly, we apply the same characterization to deep ReLU networks with whitened data and prove the same weight alignment holds. As a corollary, we prove that norm regularized deep ReLU networks yield spline interpolation for one-dimensional datasets which was previously known only for two-layer networks. Furthermore, we provide closed-form solutions for the optimal layer weights when data is rank-one or whitened. We then verify our theory via numerical experiments.

## 1 Introduction

Deep neural networks (DNNs) have become extremely popular due to their success in machine learning applications. Even though DNNs are highly over-parameterized and non-convex, simple first-order algorithms, e.g., Stochastic Gradient Descent (SGD), can be used to successfully train them. Moreover, recent work has shown that highly over-parameterized networks trained with SGD obtain simple solutions that generalize well (Savarese et al., 2019; Parhi & Nowak, 2019; Ergen & Pilanci, 2020a;b), where two-layer ReLU networks with the minimum Euclidean norm solution and zero training error are proven to fit a linear spline model in 1D regression. Therefore, regularizing the solution towards smaller norm weights might be the key to understand the generalization properties of DNNs. However, analyzing DNNs is still theoretically elusive even in the absence of nonlinear activations. Therefore, we study norm regularized DNNs and develop a framework based on convex duality such that a set of optimal solutions to the training problem can be analytically characterized.

Deep linear networks have been the subject of extensive theoretical analysis due to their tractability. A line of research (Saxe et al., 2013; Arora et al., 2018a; Laurent & Brecht, 2018; Du & Hu, 2019; Shamir, 2018) focused on GD training dynamics, however, they lack the analysis of generalization properties of deep networks. Another line of research (Gunasekar et al., 2017; Arora et al., 2019; Bhojanapalli et al., 2016) studied the generalization properties via matrix factorization and showed that linear networks trained with GD converge to minimum nuclear norm solutions. Later on, Arora et al. (2018b); Du et al. (2018) showed that gradient flow enforces the layer weights to align. Ji & Telgarsky (2019) further proved that each layer weight matrix is asymptotically rank-one. These results provide insights to characterize the structure of the optimal layer weights, however, they require multiple strong assumptions, e.g., linearly separable training data and strictly decreasing loss function, which makes the results impractical. Furthermore, Zhang et al. (2019) provided some characterizations for nonstandard networks, which are valid for hinge loss and specific regularizations where the data matrix is included. Unlike these studies, we introduce a complete characterization for the regularized deep network training problem without requiring such assumptions.

**Our contributions: 1)** We introduce a convex analytic framework that characterizes a set of optimal solutions to regularized training problems as the extreme points of a convex set, which is valid for vector outputs and popular loss functions including squared, cross entropy and hinge loss[1]; **2)** For deep linear networks with $K$ outputs, we prove that each optimal layer weight matrix aligns

---

[1] Extensions to other loss functions, e.g., cross entropy and hinge loss, are presented Appendix A.1

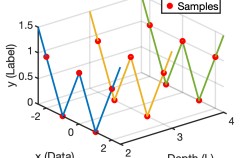

| | Width ($m$) | Depth ($L$) | Vector outputs ($K$) |
|---|---|---|---|
| Savarese et al. (2019) | $\infty$ | 2 | ✗ ($K=1$) |
| Parhi & Nowak (2019) | $\infty$ | 2 | ✗ ($K=1$) |
| Ergen & Pilanci (2020a;b) | finite | 2 | ✗ ($K=1$) |
| **Our work** | finite | $L \geq 2$ | ✓ ($K \geq 1$) |

Figure 1 & Table 1: One dimensional interpolation using $L$-layer ReLU networks with 20 neurons in each hidden layer. As predicted by Corollary 4.2, the optimal solution is given by piecewise linear splines for any $L \geq 2$. Additionally, we provide a comparison with previous studies about this characterization.

with the previous layers and becomes rank-$K$ via convex duality; **3)** For deep ReLU networks, we obtain the same weight alignment result for whitened or rank-one data matrices. **As a corollary, we achieve closed-form solutions for the optimal hidden layer weights when data is whitened or rank-one (see Theorem 4.1 and 4.3)**. As another corollary, **we prove that the optimal networks are linear spline interpolators for one-dimensional, i.e., rank-one, data which generalizes the two-layer results for one-dimensional data in Savarese et al. (2019); Parhi & Nowak (2019); Ergen & Pilanci (2020a;b) to arbitrary depth.** We note that the analysis of ReLU networks for the one dimensional data considered in these works is non-trivial, which is a special case of our rank-one/whitened data assumption.

**Notation:** We denote matrices/vectors as uppercase/lowercase bold letters. We use $\mathbf{0}_k$ (or $\mathbf{1}_k$) and $\mathbf{I}_k$ to denote a vector of zeros (or ones) and the identity matrix of size $k$, respectively. We denote the set of integers from 1 to $n$ as $[n]$. To denote Frobenius, operator, and nuclear norms, we use $\|\cdot\|_F$, $\|\cdot\|_2$, and $\|\cdot\|_*$, respectively. Furthermore, $\sigma_{max}(\cdot)$ and $\sigma_{min}(\cdot)$ represent the maximum and minimum singular values, respectively and $\mathcal{B}_2$ is defined as $\mathcal{B}_2 := \{\mathbf{u} \in \mathbb{R}^d \,|\, \|\mathbf{u}\|_2 \leq 1\}$.

## 1.1 OVERVIEW OF OUR RESULTS

We consider an $L$-layer network with layer weights $\mathbf{W}_l \in \mathbb{R}^{m_{l-1} \times m_l}$, $\forall l \in [L]$, where $m_0 = d$ and $m_L = 1$, respectively. Then, given a data matrix $\mathbf{X} \in \mathbb{R}^{n \times d}$, the output is $f_{\theta,L}(\mathbf{X}) = \mathbf{A}_{L-1}\mathbf{w}_L$, $\mathbf{A}_l = g(\mathbf{A}_{l-1}\mathbf{W}_l) \,\forall l \in [L-1]$, where $\mathbf{A}_0 = \mathbf{X}$ and $g(\cdot)$ is the activation function. Given a label vector $\mathbf{y} \in \mathbb{R}^n$, training problem can be formulated as follows

$$\min_{\{\theta_l\}_{l=1}^L} \mathcal{L}(f_{\theta,L}(\mathbf{X}), \mathbf{y}) + \beta \mathcal{R}(\theta), \tag{1}$$

where $\mathcal{L}(\cdot, \cdot)$ is an arbitrary loss function, $\mathcal{R}(\theta)$ is regularization for the layer weights, $\beta > 0$ is a regularization parameter, $\theta_l = \{\mathbf{W}_l, m_l\}$, and $\theta := \{\theta_l\}_{l=1}^L$. In the paper, for the sake of presentation simplicity, we illustrate the conventional training setup with squared loss and $\ell_2$-norm regularization, i.e., $\mathcal{L}(f_{\theta,L}(\mathbf{X}), \mathbf{y}) = \|f_{\theta,L}(\mathbf{X}) - \mathbf{y}\|_2^2$ and $\mathcal{R}(\theta) = \sum_{l=1}^L \|\mathbf{W}_l\|_F^2$. However, our analysis is valid for arbitrary loss functions and different regularization terms as proven in Appendix. Thus, we consider the following optimization problem

$$P^* = \min_{\{\theta_l\}_{l=1}^L} \mathcal{L}(f_{\theta,L}(\mathbf{X}), \mathbf{y}) + \beta \sum_{l=1}^L \|\mathbf{W}_l\|_F^2. \tag{2}$$

Next, we show that the minimum $\ell_2^2$ norm is equivalent to minimum $\ell_1$ norm after a rescaling.

**Lemma 1.1.** *The following problems are equivalent :*

$$\min_{\{\theta_l\}_{l=1}^L} \mathcal{L}(f_{\theta,L}(\mathbf{X}), \mathbf{y}) + \beta \sum_{l=1}^L \|\mathbf{W}_l\|_F^2 = \begin{array}{c} \min_{\{\theta_l\}_{l=1}^L, t} \mathcal{L}(f_{\theta,L}(\mathbf{X}), \mathbf{y}) + 2\beta\|\mathbf{w}_L\|_1 + \beta(L-2)t^2 \\ s.t. \ \mathbf{w}_{L-1,j} \in \mathcal{B}_2, \|\mathbf{W}_l\|_F \leq t, \ \forall l \in [L-2] \end{array},$$

*where $\mathbf{w}_{L-1,j}$ denotes the $j^{th}$ column of $\mathbf{W}_{L-1}$.*

Using Lemma 1.1[2], we first take the dual with respect to the output layer weights $\mathbf{w}_L$ and then change the order of min-max to achieve the following dual deep network training problem, which provides a lower bound [3]

$$P^* \geq D^* = \min_t \max_{\boldsymbol{\lambda}} \min_{\substack{\mathbf{w}_{L-1,j} \in \mathcal{B}_2, \forall j \\ \|\mathbf{W}_l\|_F \leq t, \, \forall l \in [L-2]}} -\mathcal{L}^*(\boldsymbol{\lambda}) + \beta(L-2)t^2 \text{ s.t. } \|\mathbf{A}_{L-1}^T\boldsymbol{\lambda}\|_\infty \leq 2\beta.$$

---

[2]The proof is presented in Appendix A.3.

[3]For the definitions and details see Appendix A.1.

To the best of our knowledge, the above dual deep network characterization is novel. Using this result, we first characterize a set of weights that minimize the objective via the optimality conditions and active constraints in the dual objective. We then prove the optimality of these weights by proving strong duality, i.e., $P^* = D^*$, for deep networks. We then show that, for deep linear networks with $K$ outputs, optimal weight matrices are rank-$K$ and align with the previous layers.

More importantly, the same analysis and conclusions also apply to deep ReLU networks with $K$ outputs when the input is whitened and/or rank-one. To the best of our knowledge, this is the first work providing a complete characterization for deep ReLU networks via convex duality. Based on this analysis, we even obtain closed-form solutions for the optimal layer weights. As a corollary, we show that deep ReLU networks fit a linear spline interpolation when the input is a one-dimensional dataset. We also provide an experiment in Figure 1 to verify this claim. We emphasize that this result was previously known only for two-layer networks (Savarese et al., 2019; Parhi & Nowak, 2019; Ergen & Pilanci, 2020a;b) and here we extend it to arbitrary depth $L$ (see Table 1 for details).

## 2 WARMUP: TWO-LAYER LINEAR NETWORKS

To illustrate an application of the convex dual $D^*$, we consider the simple case of two-layer linear networks with the output $f_{\theta,2}(\mathbf{X}) = \mathbf{X}\mathbf{W}_1\mathbf{w}_2$ and define the parameter space as $\theta \in \Theta = \{(\mathbf{W}_1, \mathbf{w}_2, m) \,|\, \mathbf{W}_1 \in \mathbb{R}^{d \times m}, \mathbf{w}_2 \in \mathbb{R}^m, m \in \mathbb{Z}_+\}$. Motivated by recent results (Neyshabur et al., 2014; Chizat & Bach, 2018; Savarese et al., 2019; Parhi & Nowak, 2019; Ergen & Pilanci, 2020a;b), we first focus on a minimum norm[4] variant of equation 1 when $\mathcal{L}(f_{\theta,L}(\mathbf{X}), \mathbf{y}) = \|f_{\theta,L}(\mathbf{X}) - \mathbf{y}\|_2^2$ and then extend it to equation 1. The minimum norm primal training problem can be written as

$$\min_{\theta \in \Theta} \|\mathbf{W}_1\|_F^2 + \|\mathbf{w}_2\|_2^2 \text{ s.t. } f_{\theta,2}(\mathbf{X}) = \mathbf{y}. \tag{3}$$

Using Lemma A.1[5], we equivalently have

$$P^* = \min_{\theta \in \Theta} \|\mathbf{w}_2\|_1 \text{ s.t. } f_{\theta,2}(\mathbf{X}) = \mathbf{y}, \mathbf{w}_{1,j} \in \mathcal{B}_2, \forall j, \tag{4}$$

which has the following dual form.

**Theorem 2.1.** *The dual of the problem in equation 4 is given by*

$$P^* \geq D^* = \max_{\boldsymbol{\lambda} \in \mathbb{R}^n} \boldsymbol{\lambda}^T \mathbf{y} \text{ s.t. } \max_{\mathbf{w}_1 \in \mathcal{B}_2} |\boldsymbol{\lambda}^T \mathbf{X}\mathbf{w}_1| \leq 1. \tag{5}$$

*For finite width networks, there exists a finite $m$ such that strong duality holds, i.e., $P^* = D^*$, and an optimal $\mathbf{W}_1$ for equation 4 satisfies $\|(\mathbf{X}\mathbf{W}_1^*)^T \boldsymbol{\lambda}^*\|_\infty = 1$, where $\boldsymbol{\lambda}^*$ is the dual optimal parameter.*

Using Theorem 2.1, we now characterize the optimal neurons as the extreme points of a convex set.

**Corollary 2.1.** *Theorem 2.1 implies that the optimal neurons are extreme points which solve the following problem* $\arg\max_{\mathbf{w}_1 \in \mathcal{B}_2} |\boldsymbol{\lambda}^{*T} \mathbf{X}\mathbf{w}_1|$.

**Definition 1.** We call the maximizers of the constraint in Corollary 2.1 *extreme points*.

From Theorem 2.1, we have the following dual problem

$$\max_{\boldsymbol{\lambda}} \boldsymbol{\lambda}^T \mathbf{y} \text{ s.t. } \max_{\mathbf{w}_1 \in \mathcal{B}_2} |\boldsymbol{\lambda}^T \mathbf{X}\mathbf{w}_1| \leq 1. \tag{6}$$

Let $\mathbf{X} = \mathbf{U}_x \boldsymbol{\Sigma}_x \mathbf{V}_x^T$ be the singular value decomposition (SVD) of $\mathbf{X}$[6]. If we assume that there exists $\mathbf{w}^*$ such that $\mathbf{X}\mathbf{w}^* = \mathbf{y}$ due to Proposition 2.1, then equation 6 is equivalent to

$$\max_{\tilde{\boldsymbol{\lambda}}} \tilde{\boldsymbol{\lambda}}^T \boldsymbol{\Sigma}_x \tilde{\mathbf{w}}^* \text{ s.t. } \|\boldsymbol{\Sigma}_x^T \tilde{\boldsymbol{\lambda}}\|_2 \leq 1, \tag{7}$$

where $\tilde{\boldsymbol{\lambda}} = \mathbf{U}_x^T \boldsymbol{\lambda}$ and $\tilde{\mathbf{w}}^* = \mathbf{V}_x^T \mathbf{w}^*$. Notice that in equation 7, we use an alternative formulation for the constraint, i.e., $\|\mathbf{X}^T \boldsymbol{\lambda}\|_2 \leq 1$ instead of $|\boldsymbol{\lambda}^T \mathbf{X}\mathbf{w}_1| \leq 1$, $\forall \mathbf{w}_1 \in \mathcal{B}_2$ since the extreme point is achieved when $\mathbf{w}_1 = \mathbf{X}^T \boldsymbol{\lambda}/\|\mathbf{X}^T \boldsymbol{\lambda}\|_2$. Given $\text{rank}(\mathbf{X}) = r \leq \min\{n, d\}$, we have

$$\tilde{\boldsymbol{\lambda}}^T \boldsymbol{\Sigma}_x \tilde{\mathbf{w}}^* = \tilde{\boldsymbol{\lambda}}^T \boldsymbol{\Sigma}_x \underbrace{\begin{bmatrix} \mathbf{I}_r & \mathbf{0}_{r \times d-r} \\ \mathbf{0}_{d-r \times r} & \mathbf{0}_{d-r \times d-r} \end{bmatrix} \tilde{\mathbf{w}}^*}_{\mathbf{w}_r^*} \leq \|\boldsymbol{\Sigma}_x^T \tilde{\boldsymbol{\lambda}}\|_2 \|\tilde{\mathbf{w}}_r^*\|_2 \leq \|\tilde{\mathbf{w}}_r^*\|_2, \tag{8}$$

---

[4]This corresponds to weak regularization, i.e., $\beta \to 0$ in equation 1 (see e.g. Wei et al. (2018).).

[5]All the equivalence lemmas and proofs are presented in Appendix A.3.

[6]In this paper, we use full SVD unless otherwise stated.

which shows that the maximum objective value is achieved when $\boldsymbol{\Sigma}_x^T \tilde{\boldsymbol{\lambda}} = c_1 \tilde{\mathbf{w}}_r^*$. Thus, we have

$$\mathbf{w}_1^* = \frac{\mathbf{V}_x \boldsymbol{\Sigma}_x^T \tilde{\boldsymbol{\lambda}}}{\|\mathbf{V}_x \boldsymbol{\Sigma}_x^T \tilde{\boldsymbol{\lambda}}\|_2} = \frac{\mathbf{V}_x \tilde{\mathbf{w}}_r^*}{\|\tilde{\mathbf{w}}_r^*\|_2} = \frac{\mathcal{P}_{\mathbf{X}^T}(\mathbf{w}^*)}{\|\mathcal{P}_{\mathbf{X}^T}(\mathbf{w}^*)\|_2},$$

where $\mathcal{P}_{\mathbf{X}^T}(\cdot)$ projects its input onto the range of $\mathbf{X}^T$. In the following results, we show that one can consider a planted model without loss of generality and prove strong duality for equation 4.

**Proposition 2.1.** *[Du & Hu (2019)] Given* $\mathbf{w}^* = \arg\min_{\mathbf{w}} \|\mathbf{X}\mathbf{w} - \mathbf{y}\|_2$, *we have*

$$\underset{\mathbf{W}_1, \mathbf{w}_2}{\arg\min} \|\mathbf{X}\mathbf{W}_1\mathbf{w}_2 - \mathbf{X}\mathbf{w}^*\|_2^2 = \underset{\mathbf{W}_1, \mathbf{w}_2}{\arg\min} \|\mathbf{X}\mathbf{W}_1\mathbf{w}_2 - \mathbf{y}\|_2^2.$$

**Theorem 2.2.** *Let* $\{\mathbf{X}, \mathbf{y}\}$ *be feasible for equation 4, then strong duality holds for finite width networks.*

## 2.1 REGULARIZED TRAINING PROBLEM

In this section, we define the regularized version of equation 4 as

$$\min_{\theta \in \Theta} \frac{1}{2}\|f_{\theta,2}(\mathbf{X}) - \mathbf{y}\|_2^2 + \beta\|\mathbf{w}_2\|_1 \quad \text{s.t. } \mathbf{w}_{1,j} \in \mathcal{B}_2, \forall j \tag{9}$$

which has the following dual form

$$\max_{\boldsymbol{\lambda}} -\frac{1}{2}\|\boldsymbol{\lambda} - \mathbf{y}\|_2^2 + \frac{1}{2}\|\mathbf{y}\|_2^2 \quad \text{s.t. } \max_{\mathbf{w}_1 \in \mathcal{B}_2} |\boldsymbol{\lambda}^T \mathbf{X}\mathbf{w}_1| \leq \beta.$$

Then, an optimal neuron needs to satisfy the condition

$$\mathbf{w}_1^* = \frac{\mathbf{X}^T \mathcal{P}_{\mathbf{X},\beta}(\mathbf{y})}{\|\mathbf{X}^T \mathcal{P}_{\mathbf{X},\beta}(\mathbf{y})\|_2}$$

where $\mathcal{P}_{\mathbf{X},\beta}(\cdot)$ projects its argument to $\{\mathbf{u} \in \mathbb{R}^n \mid \|\mathbf{X}^T\mathbf{u}\|_2 \leq \beta\}$. We now prove strong duality for equation 9.

**Theorem 2.3.** *Strong duality holds for equation 9 with finite width networks.*

## 2.2 TRAINING PROBLEM WITH VECTOR OUTPUTS

Here, the model is $f_{\theta,2}(\mathbf{X}) = \mathbf{X}\mathbf{W}_1\mathbf{W}_2$ to estimate $\mathbf{Y} \in \mathbb{R}^{n \times K}$, which can be optimized as follows

$$\min_{\theta \in \Theta} \|\mathbf{W}_1\|_F^2 + \|\mathbf{W}_2\|_F^2 \quad \text{s.t. } f_{\theta,2}(\mathbf{X}) = \mathbf{Y}. \tag{10}$$

Using Lemma A.2, we reformulate equation 10 as

$$\min_{\theta \in \Theta} \sum_{j=1}^{m} \|\mathbf{w}_{2,j}\|_2 \quad \text{s.t. } f_{\theta,2}(\mathbf{X}) = \mathbf{Y}, \mathbf{w}_{1,j} \in \mathcal{B}_2, \forall j. \tag{11}$$

which has the following dual with respect to $\mathbf{W}_2$

$$\max_{\boldsymbol{\Lambda}} \text{trace}(\boldsymbol{\Lambda}^T\mathbf{Y}) \quad \text{s.t. } \|\boldsymbol{\Lambda}^T\mathbf{X}\mathbf{w}_1\|_2 \leq 1, \ \forall \mathbf{w}_1 \in \mathcal{B}_2. \tag{12}$$

Since we can assume $\mathbf{Y} = \mathbf{X}\mathbf{W}^*$ due to Proposition 2.1, where $\mathbf{W}^* \in \mathbb{R}^{d \times K}$, we have

$$\text{trace}(\boldsymbol{\Lambda}^T\mathbf{Y}) = \text{trace}(\boldsymbol{\Lambda}^T\mathbf{X}\mathbf{W}^*) = \text{trace}(\boldsymbol{\Lambda}\mathbf{U}_x\boldsymbol{\Sigma}_x\tilde{\mathbf{W}}_r^*) \leq \sigma_{max}(\boldsymbol{\Lambda}^T\mathbf{U}_x\boldsymbol{\Sigma}_x)\left\|\tilde{\mathbf{W}}_r^*\right\|_* \leq \|\tilde{\mathbf{W}}_r^*\|_* \tag{13}$$

where $\sigma_{max}(\boldsymbol{\Lambda}^T\mathbf{X}) \leq 1$ due to equation 12 and $\tilde{\mathbf{W}}_r^* = \begin{bmatrix} \mathbf{I}_r & \mathbf{0}_{r \times d-r} \\ \mathbf{0}_{d-r \times r} & \mathbf{0}_{d-r \times d-r} \end{bmatrix} \mathbf{V}_x^T\mathbf{W}^*$. Given the

SVD of $\tilde{\mathbf{W}}_r^*$, i.e., $\mathbf{U}_w\boldsymbol{\Sigma}_w\mathbf{V}_w^T$, choosing

$$\boldsymbol{\Lambda}^T\mathbf{U}_x\boldsymbol{\Sigma}_x = \mathbf{V}_w \begin{bmatrix} \mathbf{I}_{r_w} & \mathbf{0}_{r_w \times d-r_w} \\ \mathbf{0}_{K-r_w \times r_w} & \mathbf{0}_{K-r_w \times d-r_w} \end{bmatrix} \mathbf{U}_w^T$$

achieves the upper-bound above, where $r_w = \text{rank}(\tilde{\mathbf{W}}_r^*)$. Thus, optimal neurons are a subset of the first $r_w$ right singular vectors of $\boldsymbol{\Lambda}\mathbf{X}$. Moreover, the next result shows that strong duality holds.

**Theorem 2.4.** *Let* $\{\mathbf{X}, \mathbf{Y}\}$ *be feasible for equation 11, then strong duality holds for finite width networks.*

### 2.2.1 REGULARIZED CASE

Here, we define the regularized version of equation 11 as follows

$$\min_{\theta \in \Theta} \frac{1}{2}\|f_{\theta,2}(\mathbf{X}) - \mathbf{Y}\|_F^2 + \beta \sum_{j=1}^{m} \|\mathbf{w}_{2,j}\|_2 \ \text{s.t.} \ \mathbf{w}_{1,j} \in \mathcal{B}_2, \forall j.$$

which has the following dual with respect to $\mathbf{W}_2$

$$\max_{\mathbf{\Lambda}} -\frac{1}{2}\|\mathbf{\Lambda} - \mathbf{Y}\|_F^2 + \frac{1}{2}\|\mathbf{Y}\|_F^2 \ \text{s.t.} \ \sigma_{max}(\mathbf{\Lambda}^T\mathbf{X}) \leq \beta.$$

Then, the optimal neurons are a subset of the maximal right singular vectors of $\mathcal{P}_{\mathbf{X},\beta}(\mathbf{Y})^T\mathbf{X}$, where $\mathcal{P}_{\mathbf{X},\beta}(\cdot)$ projects its input to the set $\{\mathbf{U} \in \mathbb{R}^{n \times K} \mid \sigma_{max}(\mathbf{U}^T\mathbf{X}) \leq \beta\}$.

**Remark 2.1.** *Note that the optimal neurons are the right singular vectors of $\mathcal{P}_{\mathbf{X},\beta}(\mathbf{Y})^T\mathbf{X}$ that achieve the upper-bound of the set, i.e., $\|\mathcal{P}_{\mathbf{X},\beta}(\mathbf{Y})^T\mathbf{X}\mathbf{w}_1^*\|_2 = \beta$, where $\|\mathbf{w}_1^*\|_2 = 1$. This implies that the optimal neurons satisfy $\|\mathbf{Y}^T\mathbf{X}\mathbf{w}_1^*\|_2 \geq \beta$. Therefore, the number of optimal neurons and the rank of the optimal weight matrix, i.e., $\mathbf{W}_1^*$, are determined by $\beta$.*

**Remark 2.2.** *There might exist optimal solutions other than the right singular vectors of $\mathcal{P}_{\mathbf{X},\beta}(\mathbf{Y})^T\mathbf{X}$. As an example, consider $\mathbf{u}_1$ and $\mathbf{u}_2$ as the optimal right singular vectors. Then, any $\mathbf{u} = \alpha_1\mathbf{u}_1 + \alpha_2\mathbf{u}_2$ with $\alpha_1^2 + \alpha_2^2 = 1$ also achieves the upper-bound, therefore, optimal.*

## 3 DEEP LINEAR NETWORKS[7]

We now consider an $L$-layer linear network with $f_{\theta,L}(\mathbf{X}) = \mathbf{X}\mathbf{W}_1 \dots \mathbf{w}_L$, and the training problem

$$P^* = \min_{\{\theta_l\}_{l=1}^L} \sum_{l=1}^{L} \|\mathbf{W}_l\|_F^2 \ \text{s.t.} \ f_{\theta,L}(\mathbf{X}) = \mathbf{y}. \tag{14}$$

**Proposition 3.1.** *First $L - 2$ hidden layer weight matrices in equation 14 have the same operator and Frobenius norms, i.e., $t_1 = t_2 = \dots = t_{L-2}$, where $t_l = \|\mathbf{W}_l\|_F = \|\mathbf{W}_l\|_2$, $\forall l \in [L-2]$.*

**Theorem 3.1.** *Optimal layer weights for equation 14 satisfy the following relation*

$$\mathbf{W}_l^* = \begin{cases} t^* \frac{\mathbf{V}_x \tilde{\mathbf{w}}_r^*}{\|\tilde{\mathbf{w}}_r^*\|_2}\boldsymbol{\rho}_1^T & \text{if } l = 1 \\ t^*\boldsymbol{\rho}_{l-1}\boldsymbol{\rho}_l^T & \text{if } 1 < l \leq L-2 \\ \boldsymbol{\rho}_{L-2} & \text{if } l = L-1 \end{cases},$$

*where $\|\boldsymbol{\rho}_l\|_2 = 1$, $\forall l \in [L-2]$ and $\tilde{\mathbf{w}}_r^*$ follows the definition in equation 8.*

This result clearly shows that the intra-layer weights need to satisfy an aligment condition. The next theorem shows that strong duality holds in this case.

**Theorem 3.2.** *Let $\{\mathbf{X}, \mathbf{y}\}$ be feasible for equation 14, then strong duality holds for finite width networks.*

**Corollary 3.1.** *Theorem 3.1 implies that deep linear networks can obtain a scaled version of $\mathbf{y}$ using only the first layer, i.e., $\mathbf{X}\mathbf{W}_1\boldsymbol{\rho}_1 = c\mathbf{y}$, where $c > 0$. Therefore, the remaining layers do not contribute to the expressive power.*

### 3.1 TRAINING PROBLEM WITH VECTOR OUTPUTS

Here, we consider vector output, i.e., $m_L = K$, deep networks with the output $f_{\theta,L}(\mathbf{X}) = \mathbf{X}\mathbf{W}_1 \dots \mathbf{W}_L$. In this case, we have the following training problem

$$\min_{\{\theta_l\}_{l=1}^L} \sum_{l=1}^{L} \|\mathbf{W}_l\|_F^2 \ \text{s.t.} \ f_{\theta,L}(\mathbf{X}) = \mathbf{Y}. \tag{15}$$

With the same approach, the optimal layer weights for equation 15 can be characterized as follows.

---

[7]Since the derivations are similar, we present the details in Appendix A.4 and A.6.

**Theorem 3.3.** *Optimal layer weight for equation 15 can be formulated as follows*

$$
\mathbf{W}_l^* = \begin{cases} t^* \sum_{j=1}^K \tilde{\mathbf{v}}_{w,j} \boldsymbol{\rho}_{1,j}^T & \text{if } l = 1 \\ t^* \sum_{j=1}^K \boldsymbol{\rho}_{l-1,j} \boldsymbol{\rho}_{l,j}^T & \text{if } 1 < l \le L - 2 \\ \sum_{j=1}^K \boldsymbol{\rho}_{L-2,j} & \text{if } l = L - 1 \end{cases},
$$

*where $\tilde{\mathbf{v}}_{w,j}$ is the $j^{th}$ maximal right singular vector of $\mathbf{\Lambda}^T \mathbf{X}$ and we may pick a set of unit norm vectors $\{\boldsymbol{\rho}_{l,j}\}_{l=1}^{L-2}$ such that $\boldsymbol{\rho}_{l,j}^T \boldsymbol{\rho}_{l,k} = 0, \ \forall j \ne k$.*

The next theorem formally proves that strong duality holds for the primal problem in equation 15.

**Theorem 3.4.** *Let $\{\mathbf{X}, \mathbf{y}\}$ be feasible for equation 15, then strong duality holds for finite width networks.*

# 4 DEEP RELU NETWORKS

Here, we consider an $L$-layer ReLU network with $f_{\theta,L}(\mathbf{X}) = \mathbf{A}_{L-1}\mathbf{w}_L$, where $\mathbf{A}_l = (\mathbf{A}_{l-1}\mathbf{W}_l)_+, \ \forall l \in [L-1], \mathbf{A}_0 = \mathbf{X}$, and $(x)_+ = \max\{0, x\}$. Below, we first state the training problem and then present our results

$$
\min_{\{\theta_l\}_{l=1}^L} \sum_{l=1}^L \|\mathbf{W}_l\|_F^2 \ \text{s.t.} \ f_{\theta,L}(\mathbf{X}) = \mathbf{y}, \tag{16}
$$

**Theorem 4.1.** *Let $\mathbf{X}$ be a rank-one data matrix such that $\mathbf{X} = \mathbf{c}\mathbf{a}_0^T$, where $\mathbf{c} \in \mathbb{R}_+^n$ and $\mathbf{a}_0 \in \mathbb{R}^d$, then strong duality holds and the optimal weights for each layer can be formulated as follows*

$$
\mathbf{W}_l = \frac{\phi_{l-1}}{\|\phi_{l-1}\|_2} \phi_l^T, \ \forall l \in [L-2], \ \mathbf{w}_{L-1} = \frac{\phi_{L-2}}{\|\phi_{L-2}\|_2},
$$

*where $\phi_0 = \mathbf{a}_0$ and $\{\phi_l\}_{l=1}^{L-2}$ is a set of vectors such that $\phi_l \in \mathbb{R}_+^{m_l}$ and $\|\phi_l\|_2 = t^*, \ \forall l \in [L-2]$.*

Our derivations can also be extended to cases with bias term. Below, we first examine a two-layer ReLU network training problem with bias term and then extend this result to a multi-layer network.

**Theorem 4.2.** *Let $\mathbf{X}$ be a data matrix such that $\mathbf{X} = \mathbf{c}\mathbf{a}_0^T$, where $\mathbf{c} \in \mathbb{R}^n$ and $\mathbf{a}_0 \in \mathbb{R}^d$. Then, a set of optimal solutions to equation 16 satisfies $\{(\mathbf{w}_i, b_i)\}_{i=1}^m$, where $\mathbf{w}_i = s_i \frac{\mathbf{a}_0}{\|\mathbf{a}_0\|_2}, b_i = -s_i c_i \|\mathbf{a}_0\|_2$ with $s_i = \pm 1, \forall i \in [m]$.*

**Corollary 4.1.** *As a result of Theorem 4.2, when we have one dimensional data, i.e., $\mathbf{x} \in \mathbb{R}^n$, an optimal solution to equation 16 can be formulated as $\{(w_i, b_i)\}_{i=1}^m$, where $\mathbf{w}_i = s_i, \ b_i = -s_i x_i$ with $s_i = \pm 1, \forall i \in [m]$. Therefore, the optimal network output has kinks only at the input data points, i.e., the output function is in the following form: $f_{\theta,2}(\hat{x}) = \sum_i (\hat{x} - x_i)_+$. Therefore, the network output becomes linear spline interpolation for one dimensional datasets.*

We now extend the results in Theorem 4.2 and Corollary 4.1 for multi-layer ReLU networks.

**Proposition 4.1.** *Theorem 4.1 still holds when we add a bias term to the last hidden layer, i.e., the output becomes $(\mathbf{A}_{L-2}\mathbf{W}_{L-1} + \mathbf{1}_n \mathbf{b}^T)_+ \mathbf{w}_L = \mathbf{y}$, where $\mathbf{A}_l = (\mathbf{A}_{l-1}\mathbf{W}_l)_+, \ \forall l \in [L-2]$.*

**Corollary 4.2.** *As a result of Theorem 4.2 and Proposition 4.1, when we have one dimensional data, i.e., $\mathbf{x} \in \mathbb{R}^n$, the optimal network output has kinks only at the input data points, i.e., the output function is in the following form: $f_{\theta,L}(\hat{x}) = \sum_i (\hat{x} - x_i)_+$. Therefore, the network output becomes linear spline interpolation for one dimensional datasets.*

**Remark 4.1.** *Note that in Corollary 4.1 and 4.2, we prove that the optimal output function for multi-layer networks are linear spline interpolators for rank-one data, which generalizes the two-layer results for one-dimensional data in Savarese et al. (2019); Parhi & Nowak (2019); Ergen & Pilanci (2020a;b) to arbitrary depth. We also remark that the analysis of ReLU networks for the one dimensional data considered in these works is non-trivial, which is a special case of our rank-one data assumption.*

The analysis in Theorem 4.1 also holds for vector output multi-layer ReLU networks as shown in the next result.

**Proposition 4.2.** *Strong duality also holds for deep ReLU networks with vector outputs and the optimal layer weights can be formulated as in Theorem 4.1.*

Now, we extend our characterization to arbitrary rank whitened data matrices and fully characterize the optimal layer weights of a deep ReLU network with $K$ outputs.

**Theorem 4.3.** *Let $\{\mathbf{X}, \mathbf{Y}\}$ be a dataset such that $\mathbf{X}\mathbf{X}^T = \mathbf{I}_n$[8] and $\mathbf{Y}$ has orthogonal columns, then the optimal weight matrices for each layer can be formulated as follows*

$$\mathbf{W}_l = \frac{1}{\sqrt{2K}} \sum_{r=1}^{2K} \frac{\phi_{l-1,r}}{\|\phi_{l-1,r}\|_2} \phi_{l,r}^T, \ \forall l \in [L-2], \ \mathbf{W}_{L-1} = \frac{1}{\sqrt{2K}} \left[ \frac{\phi_{L-2,1}}{\|\phi_{L-2,1}\|_2} \quad \cdots \quad \frac{\phi_{L-2,2K}}{\|\phi_{L-2,2K}\|_2} \right],$$

*where $(\phi_{0,2j-1}, \phi_{0,2j}) = \left( \mathbf{X}^T (\mathbf{y}_j)_+, \mathbf{X}^T (-\mathbf{y}_j)_+ \right), \ \forall j \in [K]$ and $\{\phi_{l,r}\}_{l=1}^{L-2}$ is a set of vectors such that $\phi_{l,r} \in \mathbb{R}_+^{m_l}$, $\|\phi_{l,r}\|_2 = t^*$, and $\phi_{l,i}^T \phi_{l,j} = 0, \ \forall i \neq j$.*

**Remark 4.2.** *In one hot encoded labeling, which is the conventional labeling for classification tasks, the label matrix $\mathbf{Y} \in \mathbb{R}^{n \times K}$ has nonoverlapping, therefore orthogonal, columns. Hence, classification tasks with one hot encoded labels directly satisfy the assumption in Theorem 4.3.*

**Remark 4.3.** *We note that the whitening assumption $\mathbf{X}\mathbf{X}^T = \mathbf{I}_n$ necessitates that $n \leq d$, which might appear to be restrictive. However, this case is common in few-shot classification problems with limited labels (Chen et al., 2018). Moreover, it is challenging to obtain reliable labels in problems involving high dimensional data such as in medical imaging (Hyun et al., 2020) and genetics (Singh & Yamada, 2020), where $n \leq d$ is typical. More importantly, SGD employed in deep learning frameworks, e.g., PyTorch and Tensorflow, operate in minibatches rather than the full dataset. Therefore, even when $n > d$, each gradient descent update can only be evaluated on small batches, where the batch size $n_b$ satisfies $n_b \ll d$. Hence, the $n \leq d$ case implicitly occur during the training phase.*

We note that these results also hold for regularized ReLU networks as in the previous sections and we can obtain closed-form solutions for all the layers weights as proven in the next result.

**Theorem 4.4.** *Let $\{\mathbf{X}, \mathbf{Y}\}$ be a dataset such that $\mathbf{X}^T\mathbf{X} = \mathbf{I}_n$ and $\mathbf{Y}$ has orthogonal columns, then a set of optimal layer weight matrices for the following regularized training problem*

$$\min_{\theta \in \Theta} \frac{1}{2} \|f_{\theta,L}(\mathbf{X}) - \mathbf{Y}\|_F^2 + \frac{\beta}{2} \sum_{l=1}^{L} \|\mathbf{W}_l\|_F^2 \tag{17}$$

*can be formulated as follows*

$$\mathbf{W}_l = \begin{cases} \sum_{r=1}^{2K} \frac{\phi_{l-1,r}}{\|\phi_{l-1,r}\|_2} \phi_{l,r}^T, & \text{if } 1 \leq l \leq L-1 \\ \sum_{r=1}^{2K} (\|\phi_{0,r}\|_2 - \beta) \phi_{l-1,r} \hat{\mathbf{e}}_r^T & \text{if } l = L \end{cases},$$

*where $\hat{\mathbf{e}}_{2j-1} = \hat{\mathbf{e}}_{2j} = \mathbf{e}_j, \ \forall j \in [K]$, $\mathbf{e}_j$ is the $j^{th}$ ordinary basis vector, and the other definitions follows from Theorem 4.3 except $t^* = 1$.*

**Remark 4.4.** *Theorem 4.4 proves that when the data matrix is whitened and the label matrix satisfies certain conditions, all the layer weights can be obtained as closed-form analytical formulas. We further note the conditions in this theorem are common in some generic regression/classification frameworks. As an example, for image classification tasks, it has been shown that whitening significantly improves the classification accuracy of the state-of-the-art architectures, e.g., ResNets, on benchmark datasets such as CIFAR-100 and ImageNet (Huang et al., 2018). Furthermore, since the label matrix is one hot encoded in image classification tasks, it directly satisfies the condition in the theorem. Therefore, in such cases, there is no need to train a deep ReLU network in an end-to-end manner. Instead one can directly use the closed-form formulas in Theorem 4.4.*

---

[8]This can be achieved by applying batch whitening, which often improves accuracy (Huang et al., 2018).

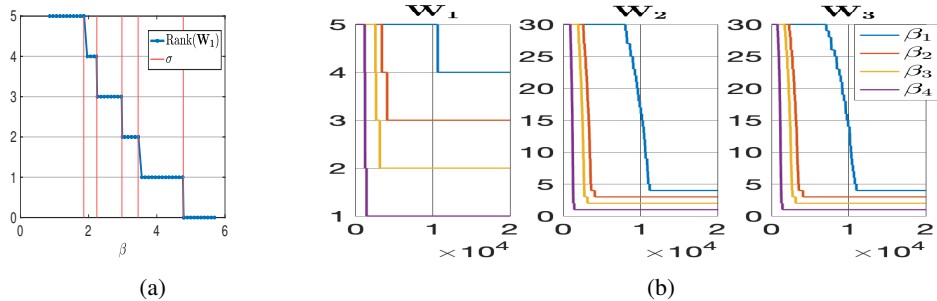

Figure 2: Verification of Remark 2.1. (a) Rank of the hidden layer weight matrix as a function of $\beta$ and (b) rank of the hidden layer weights for different regularization parameters, i.e., $\beta_1 < \beta_2 < \beta_3 < \beta_4$.

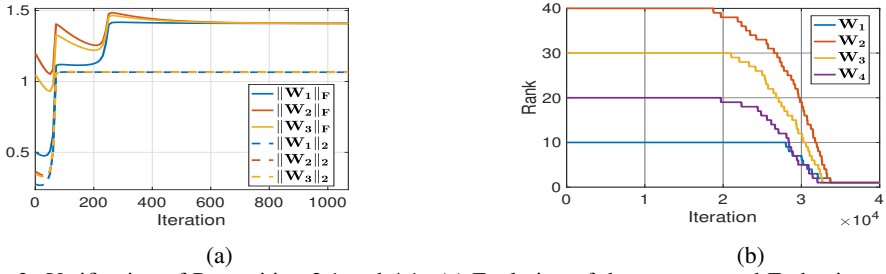

Figure 3: Verification of Proposition 3.1 and 4.1. (a) Evolution of the operator and Frobenius norms for the layer weights of a linear network and (b) Rank of the layer weights of a ReLU network with $K = 1$.

## 5 NUMERICAL EXPERIMENTS

Here, we present numerical results to verify our theoretical analysis. We first use synthetic datasets generated from a random data matrix with zero mean and identity covariance and the corresponding output vector is obtained via a randomly initialized teacher network[9]. We first consider a two-layer linear network with $\mathbf{W}_1 \in \mathbb{R}^{20 \times 50}$ and $\mathbf{W}_2 \in \mathbb{R}^{50 \times 5}$. To prove our claim in Remark 2.1, we train the network using GD with different $\beta$. In Figure 2a, we plot the rank of $\mathbf{W}_1$ as a function of $\beta$, as well as the location of the singular values of $\hat{\mathbf{W}}^* \mathbf{\Sigma}_x \mathbf{V}_x^T$ using vertical red lines. This shows that the rank of the layer changes when $\beta$ is equal to one of the singular values, which verifies Remark 2.1. We also consider a four-layer linear network with $\mathbf{W}_1 \in \mathbb{R}^{5 \times 50}$, $\mathbf{W}_2 \in \mathbb{R}^{50 \times 30}$, $\mathbf{W}_3 \in \mathbb{R}^{30 \times 40}$, and $\mathbf{W}_4 \in \mathbb{R}^{40 \times 5}$. We then select different regularization parameters as $\beta_1 < \beta_2 < \beta_3 < \beta_4$. As illustrated in Figure 2b, $\beta$ determines the rank of each weight matrix and the rank is same for all the layers, which matches with our results. Moreover, to verify Proposition 3.1, we choose $\beta$ such that the weights are rank-two. In Figure 3a, we numerically show that all the hidden layer weight matrices have the same operator and Frobenius norms. We also perform an experiment for a five-layer ReLU network with $\mathbf{W}_1 \in \mathbb{R}^{10 \times 50}$, $\mathbf{W}_2 \in \mathbb{R}^{50 \times 40}$, $\mathbf{W}_3 \in \mathbb{R}^{40 \times 30}$, $\mathbf{W}_4 \in \mathbb{R}^{30 \times 20}$, and $\mathbf{w}_5 \in \mathbb{R}^{20 \times 1}$. Here, we use data such that $\mathbf{X} = \mathbf{ca}_0^T$, where $\mathbf{c} \in \mathbb{R}_+^n$ and $\mathbf{a}_0 \in \mathbb{R}^d$. In Figure 3b, we plot the rank of each weight matrix, which converges to one as claimed Proposition 4.1.

We also verify our theory on two real benchmark datasets, i.e., MNIST (LeCun) and CIFAR10 (Krizhevsky et al., 2014). We first randomly undersample and whitened these datasets. Furthermore, we convert the labels into one hot encoded form. Then, we consider ten class classification/regression task using three multi-layer ReLU network architecture with $L = 3, 4, 5$. For each architecture, we use SGD with momentum for training and compare the training/test performance with the corresponding network constructed via the closed-form solutions (without any sort of training) in Theorem 4.3, i.e., denoted as "Theory". In Figure 4, we observe that Theory achieves the optimal training objective, which also yields smaller error and higher accuracy in the test phase. Hence, these experiments numerically verify our claims in Theorem 4.3.

## 6 CONCLUDING REMARKS

We studied regularized DNN training problems and developed an analytic framework to characterize a set of optimal solutions. We showed that optimal layer weights can be explicitly formulated as the extreme points of a convex set via the dual problem. We then proved that strong duality

---

[9]Additional numerical results can be found in Appendix A.2.

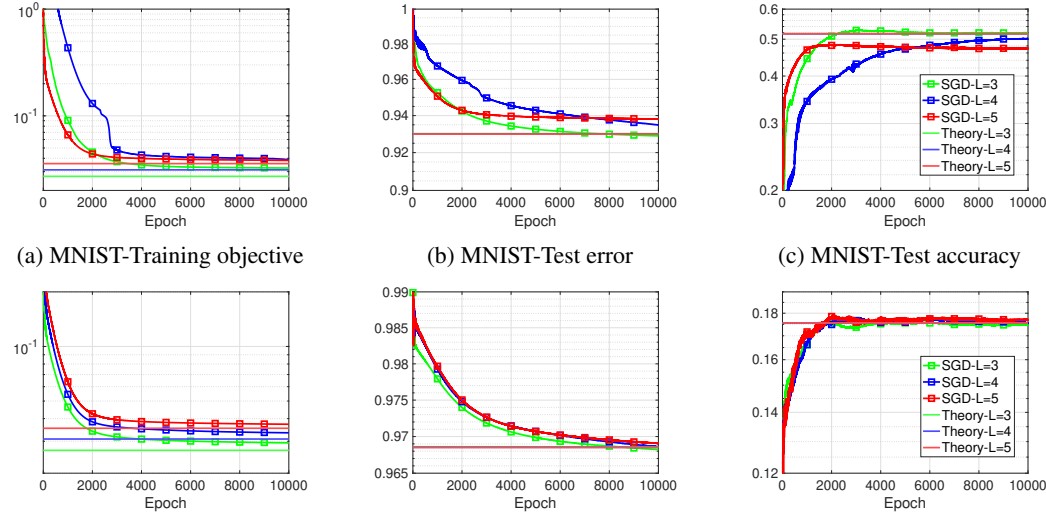

Figure 4: Training and test performance on whitened and sampled datasets, where $(n, d) = (60, 90)$, $K = 10$, $L = 3, 4, 5$ with 50 neurons per layer and we use squared loss with one hot encoding. For Theory, we use the layer weights in Theorem 4.3, which achieves the optimal performance as guaranteed by Theorem 4.3.

holds for both deep linear and ReLU networks and provided a set of optimal solutions. We also extended our derivations to the vector outputs and many other loss functions. More importantly, our analysis shows that when the input data is whitened or rank-one, instead of training an $L$-layer deep ReLU network in an end-to-end manner, one can directly use the closed-form solutions provided in Theorem 4.1, 4.3, and 4.4. As another corollary, we proved that the kinks of ReLU activations occur exactly at the input data points so that the optimized network outputs linear spline interpolations for one-dimensional datasets, which was previously known only for two-layer networks (Savarese et al., 2019; Parhi & Nowak, 2019; Ergen & Pilanci, 2020a;b). We conjecture that our extreme points characterization can also be extended to reveal the structure behind cases with arbitrary data. Therefore, one can explain the extraordinary generalization properties of DNNs.

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

# Appendix

## Table of Contents

## A  APPENDIX

Here, we present additional materials and proofs of the main results that are not included in the main paper due to the page limit. We also restate each result before the corresponding proof for the convenience of the reader.

### A.1  GENERAL LOSS FUNCTIONS

In this section, we show that our extreme point characterization holds for arbitrary convex loss functions including cross entropy and hinge loss.

$$\min_{\theta \in \Theta} \mathcal{L}(f_{\theta,2}(\mathbf{X}), \mathbf{y}) + \beta \|\mathbf{w}_2\|_1 \text{ s.t. } \mathbf{w}_{1,j} \in \mathcal{B}_2, \forall j, \tag{18}$$

where $\mathcal{L}(\cdot, \mathbf{y})$ is a convex loss function.

**Theorem A.1.** *The dual of equation 18 is given by*

$$\max_{\boldsymbol{\lambda}} -\mathcal{L}^*(\boldsymbol{\lambda}) \text{ s.t. } \|\mathbf{X}^T \boldsymbol{\lambda}\|_2 \leq \beta,$$

*where $\mathcal{L}^*$ is the Fenchel conjugate function defined as*

$$\mathcal{L}^*(\boldsymbol{\lambda}) = \max_{\mathbf{z}} \mathbf{z}^T \boldsymbol{\lambda} - \mathcal{L}(\mathbf{z}, \mathbf{y}).$$

Theorem A.1 proves that our extreme point characterization in Corollary 2.1 applies to arbitrary loss function. Therefore, optimal parameters for equation 3 and equation 9 are a subset of the same extreme point set, i.e., determined by the input data matrix $\mathbf{X}$, independent of loss function.

**Remark A.1.** *Since our characterization is generic in the sense that it holds for vector output, deep linear and deep ReLU networks (see the main paper for details), Theorem A.1 is valid for all of our derivations.*

### A.2  ADDITIONAL NUMERICAL RESULTS

Here, we present numerical results that are not included in the main paper due to the page limit. In Figure 5a, we perform an experiment to check whether the hidden neurons of a two-layer linear network align with the proposed right singular vectors. For this experiment, we select a certain $\beta$ such that $\mathbf{W}_1$ becomes rank-two. After training, we first normalize each neuron to have unit norm, i.e., $\|\mathbf{w}_{1,j}\|_2 = 1, \forall j$, and then compute the sum of the projections of each neuron onto each right singular vector, i.e., denoted as $\mathbf{v}_i$. Since we choose $\beta$ such that $\mathbf{W}_1$ is a rank-two matrix, most of the neurons align with the first two right singular vectors as expected. Therefore, this experiment verifies our analysis and claims in Remark 2.1. Furthermore, as an alternative to Figure 2a, we plot the singular values of $\mathbf{W}_1$ with respect to the regularization parameter $\beta$ in Figure 5b.

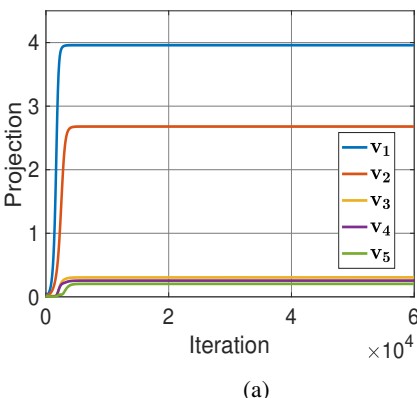 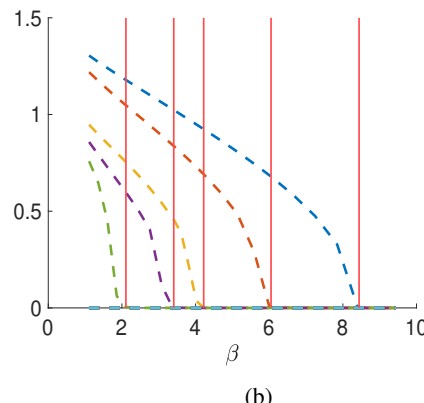

(a)                                   (b)

Figure 5: (a) Projection of the hidden neurons to the right singular vectors claimed in Remark 2.1 and (b) singular values of $\mathbf{W}_1$ with respect to $\beta$.

### A.3   EQUIVALENCE (RESCALING) LEMMAS FOR THE NON-CONVEX OBJECTIVES

In this section, we present all the equivalence (scaling transformation) lemmas we used in the main paper and the the proofs are presented in Appendix A.5, A.6, and A.7, two-layer, deep linear, and deep ReLU networks, respectively.

**Lemma 1.1.** *The following problems are equivalent :*

$$
\min_{\{\theta_l\}_{l=1}^{L}} \mathcal{L}(f_{\theta,L}(\mathbf{X}),\mathbf{y}) + \beta \sum_{l=1}^{L} \|\mathbf{W}_l\|_F^2 = 
\begin{array}{l}
\min_{\{\theta_l\}_{l=1}^{L},t} \mathcal{L}(f_{\theta,L}(\mathbf{X}),\mathbf{y}) + 2\beta\|\mathbf{w}_L\|_1 + \beta(L-2)t^2 \\
s.t. \ \mathbf{w}_{L-1,j} \in \mathcal{B}_2, \|\mathbf{W}_l\|_F \leq t, \ \forall l \in [L-2]
\end{array},
$$

*where $\mathbf{w}_{L-1,j}$ denotes the $j^{th}$ column of $\mathbf{W}_{L-1}$.*

***Proof of Lemma 1.1.*** For any $\theta \in \Theta$, we can rescale the parameters as $\bar{\mathbf{w}}_{L-1,j} = \alpha_j \mathbf{w}_{L-1,j}$ and $\bar{\mathbf{w}}_{L,j} = \mathbf{w}_{L,j}/\alpha_j$, for any $\alpha_j > 0$. Then, the network output becomes

$$
f_{\bar{\theta},L}(\mathbf{X}) = \left((\mathbf{XW}_1)_+ \dots \bar{\mathbf{W}}_{L-1}\right)_+ \bar{\mathbf{w}}_L = \left((\mathbf{XW}_1)_+ \dots \mathbf{W}_{L-1}\right)_+ \mathbf{w}_L,
$$

which proves $f_{\theta,L}(\mathbf{X}) = f_{\bar{\theta},L}(\mathbf{X})$. In addition to this, we have the following basic inequality

$$
\sum_{l=1}^{L} \|\mathbf{W}_l\|_F^2 \geq \sum_{l=1}^{L-2} \|\mathbf{W}_l\|_F^2 + 2\sum_{j=1}^{m} |w_{L,j}| \|\mathbf{w}_{L-1,j}\|_2,
$$

where the equality is achieved with the scaling choice $\alpha_j = \left(\frac{|w_{L,j}|}{\|\mathbf{w}_{L-1,j}\|_2}\right)^{\frac{1}{2}}$ is used. Since the scaling operation does not change the right-hand side of the inequality, we can set $\|\mathbf{w}_{L-1,j}\|_2 = 1, \forall j$. Therefore, the right-hand side becomes $\|\mathbf{w}_L\|_1$.

Now, let us consider a modified version of the problem, where the unit norm equality constraint is relaxed as $\|\mathbf{w}_{L-1,j}\|_2 \leq 1$. Let us also assume that for a certain index $j$, we obtain $\|\mathbf{w}_{L-1,j}\|_2 < 1$ with $w_{L,j} \neq 0$ as an optimal solution. This shows that the unit norm inequality constraint is not active for $\mathbf{w}_{L-1,j}$, and hence removing the constraint for $\mathbf{w}_{L-1,j}$ will not change the optimal solution. However, when we remove the constraint, $\|\mathbf{w}_{L-1,j}\|_2 \to \infty$ reduces the objective value since it yields $w_{L,j} = 0$. Therefore, we have a contradiction, which proves that all the constraints that correspond to a nonzero $w_{L,j}$ must be active for an optimal solution. This also shows that replacing $\|\mathbf{w}_{L-1,j}\|_2 = 1$ with $\|\mathbf{w}_{L-1,j}\|_2 \leq 1$ does not change the solution to the problem.

Then, we use the epigraph form for the norm of the first $L-2$ to achieve the equivalence. $\qquad\square$

**Lemma A.1.** *[Neyshabur et al. (2014); Savarese et al. (2019); Ergen & Pilanci (2020a;b)] The following two problems are equivalent:*

$$
\begin{array}{l}
\min_{\theta \in \Theta} \|\mathbf{W}_1\|_F^2 + \|\mathbf{w}_2\|_2^2 \\
s.t. \ f_{\theta,2}(\mathbf{X}) = \mathbf{y}
\end{array}
\quad = \quad
\begin{array}{l}
\min_{\theta \in \Theta} \|\mathbf{w}_2\|_1 \\
s.t. \ f_{\theta,2}(\mathbf{X}) = \mathbf{y}, \mathbf{w}_{1,j} \in \mathcal{B}_2
\end{array}.
$$

**Lemma A.2.** *The following problems are equivalent:*

$$\min_{\theta \in \Theta} \|\mathbf{W}_1\|_F^2 + \|\mathbf{W}_2\|_F^2 \qquad = \qquad \min_{\theta \in \Theta} \sum_{j=1}^{m} \|\mathbf{w}_{2,j}\|_2$$
$$s.t. \ f_{\theta,2}(\mathbf{X}) = \mathbf{Y} \qquad\qquad\qquad s.t. \ f_{\theta,2}(\mathbf{X}) = \mathbf{Y}, \mathbf{w}_{1,j} \in \mathcal{B}_2, \forall j$$

**Lemma A.3.** *The following problems are equivalent:*

$$\min_{\{\theta_l\}_{l=1}^{L}} \sum_{l=1}^{L} \|\mathbf{W}_l\|_F^2 \qquad = \qquad \min_{\{\theta_l\}_{l=1}^{L}, \{t_l\}_{l=1}^{L-2}} \|\mathbf{w}_L\|_1 + \sum_{l=1}^{L-2} t_l^2$$
$$s.t. \ f_{\theta,L}(\mathbf{X}) = \mathbf{y} \qquad s.t. \ f_{\theta,L}(\mathbf{X}) = \mathbf{y}, \ \mathbf{w}_{L-1,j} \in \mathcal{B}_2, \|\mathbf{W}_l\|_F \le t_l, \ \forall l \in [L-2]$$

**Lemma A.4.** *The following problems are equivalent:*

$$\min_{\{\theta_l\}_{l=1}^{L}} \sum_{l=1}^{L} \|\mathbf{W}_l\|_F^2 \qquad = \qquad \min_{\{\theta_l\}_{l=1}^{L}, \{t_l\}_{l=1}^{L-2}} \sum_{j=1}^{m_{L-1}} \|\mathbf{w}_{L,j}\|_2 + \sum_{l=1}^{L-2} t_l^2$$
$$s.t. \ f_{\theta,L}(\mathbf{X}) = \mathbf{Y} \qquad s.t. \ f_{\theta,L}(\mathbf{X}) = \mathbf{Y}, \ \mathbf{w}_{L-1,j} \in \mathcal{B}_2, \|\mathbf{W}_l\|_F \le t_l, \ \forall l \in [L-2]$$

## A.4 REGULARIZED EXTENSIONS

In this section, we present the regularized versions of the training problems presented in the main paper and the proofs are presented in Appendix A.5, A.6, and A.7, two-layer, deep linear, and deep ReLU networks, respectively.

### A.4.1 REGULARIZED TRAINING PROBLEM FOR DEEP LINEAR NETWORKS WITH SCALAR OUTPUTS

Using Lemma A.3 and Proposition 3.1, we have the following dual for the regularized version of equation 14

$$\max_{\boldsymbol{\lambda}} -\frac{1}{2}\|\boldsymbol{\lambda} - \mathbf{y}\|_2^2 \ s.t. \ \|(\mathbf{X}\mathbf{W}_1 \dots \mathbf{W}_{L-2})^T \boldsymbol{\lambda}\|_2 \le \beta, \ \forall \theta_l \in \Theta_{L-1}, \ \forall l.$$

Then, the weight matrices that maximize the value of the constraint can be described as

$$\mathbf{W}_l^* = \begin{cases} t^* \frac{\mathbf{X}^T \mathcal{P}_{\mathbf{X},\beta}(\mathbf{y})}{\|\mathbf{X}^T \mathcal{P}_{\mathbf{X},\beta}(\mathbf{y})\|_2} \boldsymbol{\rho}_1^T & \text{if } l = 1 \\ t^* \boldsymbol{\rho}_{l-1} \boldsymbol{\rho}_l^T & \text{if } 1 < l \le L-2 \\ \boldsymbol{\rho}_{L-2} & \text{if } l = L-1 \end{cases}.$$

where $\mathcal{P}_{\mathbf{X},\beta}(\cdot)$ projects its input to $\{\mathbf{u} \in \mathbb{R}^n \mid \|\mathbf{X}^T\mathbf{u}\|_2 \le \beta\gamma^{-1}\}$.

**Corollary A.1.** *The analysis above and Theorem 3.2 also show that strong duality holds for the regularized deep linear network training problem.*

### A.4.2 REGULARIZED TRAINING PROBLEM FOR DEEP LINEAR NETWORKS WITH VECTOR OUTPUT

Using Lemma A.4 and Proposition 3.1, we have the following dual for the regularized version of equation 15

$$\max_{\boldsymbol{\Lambda}} -\frac{1}{2}\|\boldsymbol{\Lambda} - \mathbf{Y}\|_F^2 \ s.t. \ \sigma_{max}(\boldsymbol{\Lambda}^T\mathbf{X}\mathbf{W}_1 \dots \mathbf{W}_{L-2}) \le \beta, \ \forall \theta_l \in \Theta_{L-1},$$

where we define $\Theta_{L-1} = \{\theta_1, \dots, \theta_{L-1} | \|\mathbf{w}_{L-1,j}\|_2 \le 1, \forall j \in [m_{L-1}], \ \|\mathbf{W}_l\|_F \le t^*, \ \forall l \in [L-2]\}$. Then, as in equation 32, a set of optimal layer weights is

$$\mathbf{W}_l^* = \begin{cases} t^* \sum_{j=1}^{K} \tilde{\mathbf{v}}_{x,j} \boldsymbol{\rho}_{1,j}^T & \text{if } l = 1 \\ t^* \sum_{j=1}^{K} \boldsymbol{\rho}_{l-1,j} \boldsymbol{\rho}_{l,j}^T & \text{if } 1 < l \le L-2 \\ \sum_{j=1}^{K} \boldsymbol{\rho}_{L-2,j} & \text{if } l = L-1 \end{cases} \tag{19}$$

where $\tilde{\mathbf{v}}_{x,j}$ is a maximal right singular vector of $\mathcal{P}_{\mathbf{X},\beta}(\mathbf{Y})^T\mathbf{X}$ and $\mathcal{P}_{\mathbf{X},\beta}(\cdot)$ projects its input to the set $\{\mathbf{U} \in \mathbb{R}^{n \times k} \mid \sigma_{max}(\mathbf{U}^T\mathbf{X}) \le \beta\gamma^{-1}\}$. Additionally, $\boldsymbol{\rho}_{l,j}$'s is an orthonormal set. Therefore, the rank of each hidden layer is determined by $\beta$ as in Remark 2.1.

A.5 PROOFS FOR THE TWO-LAYER NETWORKS

**Lemma A.1.** *[Neyshabur et al. (2014); Savarese et al. (2019); Ergen & Pilanci (2020a;b)]  The following two problems are equivalent:*

$$\min_{\theta \in \Theta} \|\mathbf{W}_1\|_F^2 + \|\mathbf{w}_2\|_2^2 \qquad\qquad \min_{\theta \in \Theta} \|\mathbf{w}_2\|_1$$
$$=$$
$$s.t.\ f_{\theta,2}(\mathbf{X}) = \mathbf{y} \qquad\qquad s.t.\ f_{\theta,2}(\mathbf{X}) = \mathbf{y}, \mathbf{w}_{1,j} \in \mathcal{B}_2$$

***Proof of Lemma A.1.*** For any $\theta \in \Theta$, we can rescale the parameters as $\bar{\mathbf{w}}_{1,j} = \alpha_j \mathbf{w}_{1,j}$ and $\bar{w}_{2,j} = w_{2,j}/\alpha_j$, for any $\alpha_j > 0$. Then, the network output becomes

$$f_{\bar{\theta},2}(\mathbf{X}) = \sum_{j=1}^{m} \bar{w}_{2,j} \mathbf{X} \bar{\mathbf{w}}_{1,j} = \sum_{j=1}^{m} \frac{w_{2,j}}{\alpha_j} \alpha_j \mathbf{X} \mathbf{w}_{1,j} = \sum_{j=1}^{m} w_{2,j} \mathbf{X} \mathbf{w}_{1,j},$$

which proves $f_{\theta,2}(\mathbf{X}) = f_{\bar{\theta},2}(\mathbf{X})$. In addition to this, we have the following basic inequality

$$\frac{1}{2} \sum_{j=1}^{m} (w_{2,j}^2 + \|\mathbf{w}_{1,j}\|_2^2) \geq \sum_{j=1}^{m} (|w_{2,j}| \, \|\mathbf{w}_{1,j}\|_2),$$

where the equality is achieved with the scaling choice $\alpha_j = \left(\frac{|w_{2,j}|}{\|\mathbf{w}_{1,j}\|_2}\right)^{\frac{1}{2}}$ is used. Since the scaling operation does not change the right-hand side of the inequality, we can set $\|\mathbf{w}_{1,j}\|_2 = 1, \forall j$. Therefore, the right-hand side becomes $\|\mathbf{w}_2\|_1$.

Now, let us consider a modified version of the problem, where the unit norm equality constraint is relaxed as $\|\mathbf{w}_{1,j}\|_2 \leq 1$. Let us also assume that for a certain index $j$, we obtain $\|\mathbf{w}_{1,j}\|_2 < 1$ with $w_{2,j} \neq 0$ as an optimal solution. This shows that the unit norm inequality constraint is not active for $\mathbf{w}_{1,j}$, and hence removing the constraint for $\mathbf{w}_{1,j}$ will not change the optimal solution. However, when we remove the constraint, $\|\mathbf{w}_{1,j}\|_2 \to \infty$ reduces the objective value since it yields $w_{2,j} = 0$. Therefore, we have a contradiction, which proves that all the constraints that correspond to a nonzero $w_{2,j}$ must be active for an optimal solution. This also shows that replacing $\|\mathbf{w}_{1,j}\|_2 = 1$ with $\|\mathbf{w}_{1,j}\|_2 \leq 1$ does not change the solution to the problem. $\qquad\square$

**Theorem 2.1.** *The dual of the problem in equation 4 is given by*

$$P^* \geq D^* = \max_{\boldsymbol{\lambda} \in \mathbb{R}^n} \boldsymbol{\lambda}^T \mathbf{y} \ \ s.t. \ \ \max_{\mathbf{w}_1 \in \mathcal{B}_2} \left| \boldsymbol{\lambda}^T \mathbf{X} \mathbf{w}_1 \right| \leq 1 \,. \tag{5}$$

*For finite width networks, there exists a finite $m$ such that strong duality holds, i.e., $P^* = D^*$, and an optimal $\mathbf{W}_1$ for equation 4 satisfies $\|(\mathbf{X}\mathbf{W}_1^*)^T \boldsymbol{\lambda}^*\|_\infty = 1$, where $\boldsymbol{\lambda}^*$ is the dual optimal parameter.*

**Corollary 2.1.** *Theorem 2.1 implies that the optimal neurons are extreme points which solve the following problem* $\arg\max_{\mathbf{w}_1 \in \mathcal{B}_2} |\boldsymbol{\lambda}^{*T} \mathbf{X} \mathbf{w}_1|$.

***Proof of Theorem 2.1 and Corollary 2.1.*** We first note that the dual of equation 4 with respect to $\mathbf{w}_2$ is

$$\min_{\theta \in \Theta \setminus \{\mathbf{w}_2\}} \max_{\boldsymbol{\lambda}} \boldsymbol{\lambda}^T \mathbf{y} \ \text{s.t.} \ \|(\mathbf{X}\mathbf{W}_1)_+^T \boldsymbol{\lambda}\|_\infty \leq 1, \ \|\mathbf{w}_{1,j}\|_2 \leq 1, \forall j.$$

Then, we can reformulate the problem as follows

$$P^* = \min_{\theta \in \Theta \setminus \{\mathbf{w}_2\}} \max_{\boldsymbol{\lambda}} \boldsymbol{\lambda}^T \mathbf{y} + \mathcal{I}(\|(\mathbf{X}\mathbf{W}_1)_+^T \boldsymbol{\lambda}\|_\infty \leq 1), \ \text{s.t.} \ \|\mathbf{w}_{1,j}\|_2 \leq 1, \forall j.$$

where $\mathcal{I}(\|(\mathbf{X}\mathbf{W}_1)^T \boldsymbol{\lambda}\|_\infty \leq 1)$ is the characteristic function of the set $\|(\mathbf{X}\mathbf{W}_1)^T \boldsymbol{\lambda}\|_\infty \leq 1$, which is defined as

$$\mathcal{I}(\|(\mathbf{X}\mathbf{W}_1)^T \boldsymbol{\lambda}\|_\infty \leq 1) = \begin{cases} 0 & \text{if } \|(\mathbf{X}\mathbf{W}_1)^T \boldsymbol{\lambda}\|_\infty \leq 1 \\ -\infty & \text{otherwise} \end{cases}.$$

Since the set $\|(\mathbf{X}\mathbf{W}_1)^T \boldsymbol{\lambda}\|_\infty \leq 1$ is closed, the function $\Phi(\boldsymbol{\lambda}, \mathbf{W}_1) = \boldsymbol{\lambda}^T \mathbf{y} + \mathcal{I}(\|(\mathbf{X}\mathbf{W}_1)^T \boldsymbol{\lambda}\|_\infty \leq 1)$ is the sum of a linear function and an upper-semicontinuous indicator function and therefore upper-semicontinuous. The constraint on $\mathbf{W}_1$ is convex and compact. We use $P^*$ to denote the

value of the above min-max program. Exchanging the order of min-max we obtain the dual problem given in equation 5, which establishes a lower bound $D^*$ for the above problem:

$$
\begin{aligned}
P^* \geq D^* &= \max_{\boldsymbol{\lambda}} \min_{\theta \in \Theta \backslash \{\mathbf{w}_2\}} \boldsymbol{\lambda}^T \mathbf{y} + \mathcal{I}(\|(\mathbf{X}\mathbf{W}_1)^T \boldsymbol{\lambda}\|_\infty \leq 1), \text{ s.t. } \|\mathbf{w}_{1,j}\|_2 \leq 1, \forall j, \\
&= \max_{\boldsymbol{\lambda}} \boldsymbol{\lambda}^T \mathbf{y}, \text{ s.t. } \|(\mathbf{X}\mathbf{W}_1)^T \boldsymbol{\lambda}\|_\infty \leq 1 \, \forall \mathbf{w}_{1,j} : \|\mathbf{w}_{1,j}\|_2 \leq 1, \forall j, \\
&= \max_{\boldsymbol{\lambda}} \boldsymbol{\lambda}^T \mathbf{y}, \text{ s.t. } \|(\mathbf{X}\mathbf{w}_1)^T \boldsymbol{\lambda}\|_\infty \leq 1 \, \forall \mathbf{w}_1 : \|\mathbf{w}_1\|_2 \leq 1,
\end{aligned}
$$

We now show that strong duality holds for infinite size NNs. The dual of the semi-infinite program in equation 5 is given by (see Section 2.2 of Goberna & López-Cerdá (1998) and also Bach (2017))

$$
\min \|\boldsymbol{\mu}\|_{TV}
$$
$$
\text{s.t.} \int_{\mathbf{w}_1 \in \mathcal{B}_2} \mathbf{X}\mathbf{w}_1 d\boldsymbol{\mu}(\mathbf{w}_1) = \mathbf{y},
$$

where TV is the total variation norm of the Radon measure $\boldsymbol{\mu}$. This expression coincides with the infinite-size NN as given in Bach (2017), and therefore strong duality holds. We also note that although the above formulation involves an infinite dimensional integral form, by Caratheodory's theorem, the integral can be represented as a finite summation of at most $n+1$ Dirac delta functions (Rosset et al., 2007). Next we invoke the semi-infinite optimality conditions for the dual problem in equation 5, in particular we apply Theorem 7.2 of Goberna & López-Cerdá (1998). We first define the set

$$
\mathbf{K} = \mathbf{cone} \left\{ \begin{pmatrix} s\mathbf{X}\mathbf{w}_1 \\ 1 \end{pmatrix}, \mathbf{w}_1 \in \mathcal{B}_2, s \in \{-1, +1\}; \begin{pmatrix} \mathbf{0}_n \\ -1 \end{pmatrix} \right\}.
$$

Note that $\mathbf{K}$ is the union of finitely many convex closed sets, since the function $\mathbf{X}\mathbf{w}_1$ can be expressed as the union of finitely many convex closed sets. Therefore the set $\mathbf{K}$ is closed. By Theorem 5.3 Goberna & López-Cerdá (1998), this implies that the set of constraints in equation 5 forms a Farkas-Minkowski system. By Theorem 8.4 of Goberna & López-Cerdá (1998), primal and dual values are equal, given that the system is consistent. Moreover, the system is discretizable, i.e., there exists a sequence of problems with finitely many constraints whose optimal values approach to the optimal value of equation 5. The optimality conditions in Theorem 7.2 Goberna & López-Cerdá (1998) implies that $\mathbf{y} = \mathbf{X}\mathbf{W}_1^* \mathbf{w}_2^*$ for some vector $\mathbf{w}_2^*$. Since the primal and dual values are equal, we have $\boldsymbol{\lambda}^{*T} \mathbf{y} = \boldsymbol{\lambda}^{*T} \mathbf{X}\mathbf{W}_1^* \mathbf{w}_2^* = \|\mathbf{w}_2^*\|_1$, which shows that the primal-dual pair $(\{\mathbf{w}_2^*, \mathbf{W}_1^*\}, \boldsymbol{\lambda}^*)$ is optimal. Thus, the optimal neuron weights $\mathbf{W}_1^*$ satisfy $\|(\mathbf{X}\mathbf{W}_1^*)^T \boldsymbol{\lambda}^*\|_\infty = 1$. $\quad\square$

**Proposition 2.1.** *[Du & Hu (2019)] Given* $\mathbf{w}^* = \arg\min_{\mathbf{w}} \|\mathbf{X}\mathbf{w} - \mathbf{y}\|_2$*, we have*

$$
\arg\min_{\mathbf{W}_1, \mathbf{w}_2} \|\mathbf{X}\mathbf{W}_1 \mathbf{w}_2 - \mathbf{X}\mathbf{w}^*\|_2^2 = \arg\min_{\mathbf{W}_1, \mathbf{w}_2} \|\mathbf{X}\mathbf{W}_1 \mathbf{w}_2 - \mathbf{y}\|_2^2.
$$

***Proof of Proposition 2.1.*** Let us first define a variable $\mathbf{w}^*$ that minimizes the following problem

$$
\mathbf{w}^* = \min_{\mathbf{w}} \|\mathbf{X}\mathbf{w} - \mathbf{y}\|_2^2.
$$

Thus, the following relation holds

$$
\mathbf{X}^T(\mathbf{X}\mathbf{w}^* - \mathbf{y}) = \mathbf{0}_d.
$$

Then, for any $\mathbf{w} \in \mathbb{R}^d$, we have

$$
\begin{aligned}
f(\mathbf{w}) &= \|\mathbf{X}\mathbf{w} - \mathbf{X}\mathbf{w}^* + \mathbf{X}\mathbf{w}^* - \mathbf{y}\|_2^2 \\
&= \|\mathbf{X}\mathbf{w} - \mathbf{X}\mathbf{w}^*\|_2^2 + 2(\mathbf{w} - \mathbf{w}^*)^T \underbrace{\mathbf{X}^T(\mathbf{X}\mathbf{w}^* - \mathbf{y})}_{=\mathbf{0}_d} + \|\mathbf{X}\mathbf{w}^* - \mathbf{y}\|_2^2 \\
&= \|\mathbf{X}\mathbf{w} - \mathbf{X}\mathbf{w}^*\|_2^2 + \|\mathbf{X}\mathbf{w}^* - \mathbf{y}\|_2^2.
\end{aligned}
$$

Notice that $\|\mathbf{X}\mathbf{w}^* - \mathbf{y}\|_2^2$ does not depend on $\mathbf{w}$, thus, the relation above proves that minimizing $f(\mathbf{w})$ is equivalent to minimizing $\|\mathbf{X}\mathbf{w} - \mathbf{X}\mathbf{w}^*\|_2^2$, where $\mathbf{w}^*$ is the planted model parameter. Therefore, the planted model assumption does not change solution to the linear network training problem in equation 4. $\quad\square$

**Theorem 2.2.** *Let $\{\mathbf{X}, \mathbf{y}\}$ be feasible for equation 4, then strong duality holds for finite width networks.*

***Proof of Theorem 2.2.*** Since there exists a single extreme point, we can construct a weight vector $\mathbf{w}_e \in \mathbb{R}^d$ that is the extreme point. Then, the dual of equation 4 with $\mathbf{W}_1 = \mathbf{w}_e$ is

$$D_e^* = \max_{\boldsymbol{\lambda}} \boldsymbol{\lambda}^T \mathbf{y} \text{ s.t. } \|(\mathbf{X}\mathbf{w}_e)^T \boldsymbol{\lambda}\|_\infty \le 1. \tag{20}$$

Then, we have

$$
\begin{aligned}
P^* = \min_{\theta \in \Theta \setminus \{\mathbf{w}_2\}} \max_{\boldsymbol{\lambda}} \boldsymbol{\lambda}^T \mathbf{y} \qquad & \ge \quad \max_{\boldsymbol{\lambda}} \min_{\theta \in \Theta \setminus \{\mathbf{w}_2\}} \boldsymbol{\lambda}^T \mathbf{y} \\
\text{s.t } \|(\mathbf{X}\mathbf{W}_1)^T \boldsymbol{\lambda}\|_\infty \le 1, \ \|\mathbf{w}_{1,j}\|_2 \le 1, \forall j \qquad & \phantom{\ge} \quad \text{s.t. } \|(\mathbf{X}\mathbf{W}_1)^T \boldsymbol{\lambda}\|_\infty \le 1, \ \|\mathbf{w}_{1,j}\|_2 \le 1, \forall j \\
& = \quad \max_{\boldsymbol{\lambda}} \boldsymbol{\lambda}^T \mathbf{y} \\
& \phantom{=} \quad \text{s.t. } \|(\mathbf{X}\mathbf{w}_e)^T \boldsymbol{\lambda}\|_\infty \le 1 \\
& = \quad D_e^* = D^* \tag{21}
\end{aligned}
$$

where the first inequality follows from changing order of min-max to obtain a lower bound and the equality in the second line follows from Corollary 2.1.

From the fact that an infinite width NN can always find a solution with the objective value lower than or equal to the objective value of a finite width NN, we have

$$
\begin{aligned}
P_e^* = \min_{\theta \in \Theta \setminus \{\mathbf{W}_{1,m}\}} |w_2| \qquad & \ge \qquad P^* \min_{\theta \in \Theta} \|\mathbf{w}_2\|_1 \tag{22} \\
\text{s.t. } \mathbf{X}\mathbf{w}_e w_2 = \mathbf{y} \qquad & \phantom{\ge} \quad \text{s.t. } \mathbf{X}\mathbf{W}_1 \mathbf{w}_2 = \mathbf{y}, \ \|\mathbf{w}_{1,j}\|_2 \le 1, \forall j,
\end{aligned}
$$

where $P^*$ is the optimal value of the original problem with infinitely many neurons. Now, notice that the optimization problem on the left hand side of equation 22 is convex since it is an $\ell_1$-norm minimization problem with linear equality constraints. Therefore, strong duality holds for this problem, i.e., $P_e^* = D_e^*$. Using this result along with equation 21, we prove that strong duality holds for a finite width NN, i.e., $P_e^* = P^* = D^* = D_e^*$.

$\square$

**Theorem 2.3.** *Strong duality holds for equation 9 with finite width networks.*

***Proof of Theorem 2.3.*** Since there exists a single extreme point, we can construct a weight vector $\mathbf{w}_e \in \mathbb{R}^d$ that is the extreme point. Then, the dual of equation 9 with $\mathbf{W}_1 = \mathbf{w}_e$

$$D_e^* = \max_{\boldsymbol{\lambda}} -\frac{1}{2}\|\boldsymbol{\lambda} - \mathbf{y}\|_2^2 + \frac{1}{2}\|\mathbf{y}\|_2^2 \text{ s.t. } |\boldsymbol{\lambda}^T \mathbf{X}\mathbf{w}_e| \le \beta.$$

Then the rest of the proof directly follows Proof of Theorem 2.2.

$\square$

**Theorem A.1.** *The dual of equation 18 is given by*

$$\max_{\boldsymbol{\lambda}} -\mathcal{L}^*(\boldsymbol{\lambda}) \text{ s.t. } \|\mathbf{X}^T \boldsymbol{\lambda}\|_2 \le \beta,$$

*where $\mathcal{L}^*$ is the Fenchel conjugate function defined as*

$$\mathcal{L}^*(\boldsymbol{\lambda}) = \max_{\mathbf{z}} \mathbf{z}^T \boldsymbol{\lambda} - \mathcal{L}(\mathbf{z}, \mathbf{y}).$$

***Proof of Theorem A.1.*** The proof follows from classical Fenchel duality (Boyd & Vandenberghe, 2004). We first describe equation 18 in an equivalent form as follows

$$\min_{\mathbf{z}, \theta \in \Theta} \mathcal{L}(\mathbf{z}, \mathbf{y}) + \beta \|\mathbf{w}_2\|_1 \text{ s.t. } \mathbf{z} = \mathbf{X}\mathbf{W}_1 \mathbf{w}_2, \ \|\mathbf{w}_{1,j}\|_2 \le 1, \forall j.$$

Then the dual function is

$$g(\boldsymbol{\lambda}) = \min_{\mathbf{z}, \theta \in \Theta} \mathcal{L}(\mathbf{z}, \mathbf{y}) - \boldsymbol{\lambda}^T \mathbf{z} + \boldsymbol{\lambda}^T \mathbf{X}\mathbf{W}_1 \mathbf{w}_2 + \beta \|\mathbf{w}_2\|_1 \text{ s.t. } \|\mathbf{w}_{1,j}\|_2 \le 1, \forall j.$$

Therefore, using the classical Fenchel duality (Boyd & Vandenberghe, 2004) yields the claimed dual form.

$\square$

**Lemma A.2.** *The following problems are equivalent:*

$$
\begin{aligned}
&\min_{\theta \in \Theta} \|\mathbf{W}_1\|_F^2 + \|\mathbf{W}_2\|_F^2 \\
&\text{s.t. } f_{\theta,2}(\mathbf{X}) = \mathbf{Y}
\end{aligned}
\qquad = \qquad
\begin{aligned}
&\min_{\theta \in \Theta} \sum_{j=1}^{m} \|\mathbf{w}_{2,j}\|_2 \\
&\text{s.t. } f_{\theta,2}(\mathbf{X}) = \mathbf{Y}, \mathbf{w}_{1,j} \in \mathcal{B}_2, \forall j
\end{aligned} .
$$

*Proof of Lemma A.2.* The proof directly follows from Proof of Lemma A.1. □

**Theorem 2.4.** *Let $\{\mathbf{X}, \mathbf{Y}\}$ be feasible for equation 11, then strong duality holds for finite width networks.*

*Proof of Theorem 2.4.* Since there exist $r_w$ possible extreme points, we can construct a weight matrix $\mathbf{W}_e \in \mathbb{R}^{d \times r_w}$ that consists of all the possible extreme points. Then, the dual of equation 11 with $\mathbf{W}_1 = \mathbf{W}_e$

$$
D_e^* = \max_{\mathbf{\Lambda}} \text{trace}(\mathbf{\Lambda}^T \mathbf{Y}) \text{ s.t. } \|\mathbf{\Lambda}^T \mathbf{X} \mathbf{w}_{e,j}\|_2 \leq 1, \forall j \in [r_w].
$$

Then the rest of the proof directly follows Proof of Theorem 2.2. □

### A.6   Proofs for the deep linear networks

**Lemma A.3.** *The following problems are equivalent:*

$$
\begin{aligned}
&\min_{\{\theta_l\}_{l=1}^{L}} \sum_{l=1}^{L} \|\mathbf{W}_l\|_F^2 \\
&\text{s.t. } f_{\theta,L}(\mathbf{X}) = \mathbf{y}
\end{aligned}
\qquad = \qquad
\begin{aligned}
&\min_{\{\theta_l\}_{l=1}^{L}, \{t_l\}_{l=1}^{L-2}} \|\mathbf{w}_L\|_1 + \sum_{l=1}^{L-2} t_l^2 \\
&\text{s.t. } f_{\theta,L}(\mathbf{X}) = \mathbf{y}, \ \mathbf{w}_{L-1,j} \in \mathcal{B}_2, \|\mathbf{W}_l\|_F \leq t_l, \ \forall l \in [L-2]
\end{aligned} .
$$

*Proof of Lemma A.3.* Applying the scaling trick in Lemma A.1 to the last two layers of the $L$-layer network in equation 14 gives

$$
\begin{aligned}
&\min_{\{\theta_l\}_{l=1}^{L}, \{t_l\}_{l=1}^{L-2}} \|\mathbf{w}_L\|_1 + \sum_{l=1}^{L-2} \|\mathbf{W}_l\|_F^2 \\
&\text{s.t. } \|\mathbf{w}_{L-1,j}\|_2 \leq 1, \forall j \in [m_{L-1}] \\
&\mathbf{X}\mathbf{W}_1 \ldots \mathbf{W}_{L-1}\mathbf{w}_L = \mathbf{y}
\end{aligned} .
$$

Then, we use the epigraph form for the norm of the first $L-2$ to achieve the equivalence. □

**Proposition 3.1.** *First $L-2$ hidden layer weight matrices in equation 14 have the same operator and Frobenius norms, i.e., $t_1 = t_2 = \ldots = t_{L-2}$, where $t_l = \|\mathbf{W}_l\|_F = \|\mathbf{W}_l\|_2, \forall l \in [L-2]$.*

*Proof of Proposition 3.1.* Let us first denote the sum of the norms for the first $L-2$ layer as $t$, i.e., $t = \sum_{l=1}^{L-2} t_l$, where $t_l = \|\mathbf{W}_l\|_2 = \|\mathbf{W}_l\|_F$ since the upper-bound is achieved when the matrices are rank-one (seeequation 28). Then, to find the extreme points, we need to solve the following problem

$$
\max_{\{\theta_l\}_{l=1}^{L-2}} \|\mathbf{W}_{L-2}\|_2 \ldots \|\mathbf{W}_1\|_2 \|\mathbf{V}_x \tilde{\mathbf{w}}_r^*\|_2 .
$$

We can equivalently rewrite this problem using the variables $\{t_l\}_{l=1}^{L-2}$ as follows

$$
\begin{aligned}
&\max_{\{t_l\}_{l=1}^{L-2}} \prod_{l=1}^{L-2} t_l \\
&\text{s.t. } t = \sum_{l=1}^{L-2} t_l, \ t_l \geq 0
\end{aligned}
\qquad = \qquad
\begin{aligned}
&\max_{\{t_l\}_{l=1}^{L-3}} \left( t - \sum_{l=1}^{L-3} t_l \right) \prod_{j=1}^{L-3} t_l \\
&\text{s.t. } \sum_{l=1}^{L-3} t_l \leq t, \ t_l \geq 0
\end{aligned} .
$$

If we take the derivative of the objective function of the latter problem, i.e., denoted as $f(t_1, \ldots, t_{L-3})$, with respect to $t_k$, we obtain the following

$$\frac{\partial f(t_1, \ldots, t_{L-3})}{\partial t_k} = t \prod_{\substack{l=1 \\ l \neq k}}^{L-3} t_l - 2 \prod_{\substack{l=1}}^{L-3} t_l - \sum_{\substack{l=1 \\ l \neq k}}^{L-3} t_l \prod_{\substack{j=1 \\ l \neq k}}^{L-3} t_l.$$

Then, equating the derivative to zero yields the following relation

$$t_k^* = t - \sum_{l=1}^{L-3} t_l^*$$

where $t_k^*$ denotes the optimal operator norm for the $k^{\text{th}}$ layer's weight matrix. We also note that these solutions satisfy the constraints in the optimization problem above. Since by definition $t - \sum_{l=1}^{L-3} t_l^* = t_{L-2}^*$, we have $t_1^* = t_2^* = \ldots = t_{L-2}^*$. $\qquad\square$

**Theorem 3.1.** *Optimal layer weights for equation 14 satisfy the following relation*

$$\mathbf{W}_l^* = \begin{cases} t^* \frac{\mathbf{V}_x \tilde{\mathbf{w}}_r^*}{\|\tilde{\mathbf{w}}_r^*\|_2} \boldsymbol{\rho}_1^T & \text{if } l = 1 \\ t^* \boldsymbol{\rho}_{l-1} \boldsymbol{\rho}_l^T & \text{if } 1 < l \leq L - 2 \\ \boldsymbol{\rho}_{L-2} & \text{if } l = L - 1 \end{cases},$$

*where* $\|\boldsymbol{\rho}_l\|_2 = 1$, $\forall l \in [L - 2]$ *and* $\tilde{\mathbf{w}}_r^*$ *follows the definition in equation 8.*

***Proof of Theorem 3.1.*** Using Lemma A.3 and Proposition 3.1, we have the following dual problem for equation 14

$$P^* = \min_{\{\theta_l\}_{l=1}^{L-1}, t} \max_{\boldsymbol{\lambda}} \boldsymbol{\lambda}^T \mathbf{y} + (L-2)t^2 \text{ s.t. } |(\mathbf{X}\mathbf{W}_1 \ldots \mathbf{w}_{L-1,j})^T \boldsymbol{\lambda}| \leq 1, \mathbf{w}_{L-1,j} \in \mathcal{B}_2 \qquad (23)$$

$$\|\mathbf{W}_l\|_F \leq t, \ \forall l \in [L-2].$$

Now, let us assume that the optimal Frobenius norm for each layer $l$ is $t^*$ [10]. Then, if we define $\Theta_{L-1} = \{\theta_1, \ldots, \theta_{L-1} | \|\mathbf{w}_{L-1,j}\|_2 \leq 1, \forall j \in [m_{L-1}], \|\mathbf{W}_l\|_F \leq t^*, \forall l \in [L-2]\}$, equation 23 reduces to the following problem

$$P^* \geq D^* = \max_{\boldsymbol{\lambda}} \boldsymbol{\lambda}^T \mathbf{y} \text{ s.t. } |(\mathbf{X}\mathbf{W}_1 \ldots \mathbf{w}_{L-1})^T \boldsymbol{\lambda}| \leq 1, \ \forall \theta_l \in \Theta_{L-1}, \ \forall l, \qquad (24)$$

where we change the order of min-max to obtain a lower bound for equation 23. The dual of the semi-infinite problem in equation 24 is given by

$$\min \|\boldsymbol{\mu}\|_{TV} \text{ s.t. } \int_{\{\theta_l\}_{l=1}^{L-1} \in \Theta_{L-1}} \mathbf{X}\mathbf{W}_1 \ldots \mathbf{w}_{L-1} d\boldsymbol{\mu}(\theta_1, \ldots, \theta_{L-1}) = \mathbf{y}, \qquad (25)$$

where $\boldsymbol{\mu}$ is a signed Radon measure and $\|\cdot\|_{TV}$ is the total variation norm. We emphasize that equation 25 has infinite width in each layer, however, an application of Caratheodory's theorem shows that the measure $\boldsymbol{\mu}$ in the integral can be represented by finitely many (at most $n + 1$) Dirac delta functions (Rosset et al., 2007). Such selection of $\boldsymbol{\mu}$ yields the following problem

$$P_m^* = \min_{\{\theta_l\}_{l=1}^L} \|\mathbf{w}_L\|_1 \text{ s.t. } \sum_{j=1}^{m_{L-1}} \mathbf{X}\mathbf{W}_1^j \ldots \mathbf{w}_{L-1}^j w_{L,j} = \mathbf{y}, \ \theta_l^j \in \Theta_{L-1}, \ \forall l \qquad (26)$$

We first note that since the model in equation 26 has multiple weight matrices for each layer, it has more expressive power than a regular network. Thus, we have $P^* \geq P_m^*$. Since the dual of equation 14 and equation 26 are the same, we also have $D_m^* = D^*$, where $D_m^*$ is the optimal dual value for equation 26.

We now apply the variable change in equation 7 to equation 24 as follows

$$\max_{\boldsymbol{\lambda}} \tilde{\boldsymbol{\lambda}}^T \boldsymbol{\Sigma}_x \tilde{\mathbf{w}}_r^* \text{ s.t. } \|\mathbf{W}_{L-2}^T \ldots \mathbf{W}_1^T \mathbf{V}_x \boldsymbol{\Sigma}_x^T \tilde{\boldsymbol{\lambda}}\|_2 \leq 1, \ \forall \theta_l \in \Theta_{L-1}, \ \forall l \qquad (27)$$

---

[10]With this assumption, $(L-2)t^2$ becomes constant so we ignore this term for the rest of our derivations.

which shows that the maximum objective value is achieved when $\boldsymbol{\Sigma}_x^T \tilde{\boldsymbol{\lambda}} = c_1 \tilde{\mathbf{w}}_r^*$. Thus, the optimal layer weights can be found as the maximizers of the constraint when $\boldsymbol{\Sigma}_x^T \tilde{\boldsymbol{\lambda}} = c_1 \tilde{\mathbf{w}}_r^*$. To find the formulations explicitly, we first find an upper-bound for the constraint in equation 27 as follows

$$\|\mathbf{W}_{L-2}^T \ldots \mathbf{W}_1^T \mathbf{V}_x \boldsymbol{\Sigma}_x^T \tilde{\boldsymbol{\lambda}}\|_2 = c_1 \|\mathbf{W}_{L-2}^T \ldots \mathbf{W}_1^T \mathbf{V}_x \tilde{\mathbf{w}}_r^*\|_2 \leq c_1 \|\mathbf{W}_{L-2}\|_2 \ldots \|\mathbf{V}_x \tilde{\mathbf{w}}_r^*\|_2 \leq c_1 \gamma \|\mathbf{V}_x \tilde{\mathbf{w}}_r^*\|_2,$$

where the last inequality follows from the constraint on each layer weight's norm and $\gamma = {t^*}^{L-2}$. This upper-bound can be achieved when the layer weights are

$$\mathbf{W}_l^* = \begin{cases} t^* \frac{\mathbf{V}_x \tilde{\mathbf{w}}_r^*}{\|\tilde{\mathbf{w}}_r^*\|_2} \boldsymbol{\rho}_1^T & \text{if } l = 1 \\ t^* \boldsymbol{\rho}_{l-1} \boldsymbol{\rho}_l^T & \text{if } 1 < l \leq L-2 \\ \boldsymbol{\rho}_{L-2} & \text{if } l = L-1 \end{cases}, \tag{28}$$

where $\|\boldsymbol{\rho}_l\|_2 = 1$, $\forall l \in [L-2]$. This shows that the weight matrices are rank-one and align with each other. Therefore, an arbitrary set of unit norm vectors, i.e., $\{\boldsymbol{\rho}_l\}_{l=1}^{L-2}$ can be chosen to achieve the maximum dual objective.

We note that the layer weights in equation 28 are optimal for the relaxed problem in equation 26. However, since there exists a single possible choice for the left singular vector of $\mathbf{W}_1$ and we can select an arbitrary set for $\{\boldsymbol{\rho}_l\}_{l=1}^{L-2}$, we achieve $D_m^* = D^*$ using the same layer weights. Therefore, the set of weights in equation 28 are also optimal for equation 14. $\qquad \square$

**Theorem 3.3.** *Optimal layer weight for equation 15 can be formulated as follows*

$$\mathbf{W}_l^* = \begin{cases} t^* \sum_{j=1}^K \tilde{\mathbf{v}}_{w,j} \boldsymbol{\rho}_{1,j}^T & \text{if } l = 1 \\ t^* \sum_{j=1}^K \boldsymbol{\rho}_{l-1,j} \boldsymbol{\rho}_{l,j}^T & \text{if } 1 < l \leq L-2 \\ \sum_{j=1}^K \boldsymbol{\rho}_{L-2,j} & \text{if } l = L-1 \end{cases},$$

*where $\tilde{\mathbf{v}}_{w,j}$ is the $j^{th}$ maximal right singular vector of $\boldsymbol{\Lambda}^T \mathbf{X}$ and we may pick a set of unit norm vectors $\{\boldsymbol{\rho}_{l,j}\}_{l=1}^{L-2}$ such that $\boldsymbol{\rho}_{l,j}^T \boldsymbol{\rho}_{l,k} = 0$, $\forall j \neq k$.*

*Proof of Theorem 3.3.* Using Proposition 3.1 and Lemma A.4, we obtain the following dual problem

$$\max_{\boldsymbol{\Lambda}} \text{trace}(\boldsymbol{\Lambda}^T \mathbf{Y}) \text{ s.t. } \sigma_{max}(\boldsymbol{\Lambda}^T \mathbf{X} \mathbf{W}_1 \ldots \mathbf{W}_{L-2}) \leq 1, \forall \theta_l \in \Theta_{L-1}. \tag{29}$$

It is straightforward to show that the optimal layer weights are the extreme points of the constraint in equation 29, which achieves the following upper-bound

$$\max_{\{\theta_l\}_{l=1}^{L-2} \in \Theta_{L-1}} \sigma_{max}(\boldsymbol{\Lambda}^T \mathbf{X} \mathbf{W}_1 \ldots \mathbf{W}_{L-2}) \leq \sigma_{max}(\boldsymbol{\Lambda}^T \mathbf{X}) \gamma.$$

This upper-bound is achieved when the first $L-2$ layer weights are rank-one with the singular value $t^*$ by Proposition 3.1. Additionally, the left singular vector of $\mathbf{W}_1$ needs to align with one of the maximum right singular vectors of $\boldsymbol{\Lambda}^T \mathbf{X}$. Since the upper-bound for the objective is achievable for any $\boldsymbol{\Lambda}$, we can maximize the objective value, as in equation 13, by choosing a matrix $\boldsymbol{\Lambda}$ such that

$$\boldsymbol{\Lambda}^T \mathbf{U}_x \boldsymbol{\Sigma}_x = \mathbf{V}_w \begin{bmatrix} \gamma^{-1} \mathbf{I}_{r_w} & \mathbf{0}_{r_x \times d-r_w} \\ \mathbf{0}_{k-r_w \times r_x} & \mathbf{0}_{k-r_w \times d-r_w} \end{bmatrix} \mathbf{U}_w^T$$

where $\mathbf{W}_r^* = \mathbf{U}_w \boldsymbol{\Sigma}_w \mathbf{V}_w^T$. Thus, a set of optimal layer weights can be formulated as follows

$$\mathbf{W}_l^j = \begin{cases} t^* \tilde{\mathbf{v}}_{w,j} \boldsymbol{\rho}_{1,j}^T & \text{if } l = 1 \\ t^* \boldsymbol{\rho}_{l-1,j} \boldsymbol{\rho}_{l,j}^T & \text{if } 1 < l \leq L-2 \\ \boldsymbol{\rho}_{L-2,j} & \text{if } l = L-1 \end{cases}, \tag{30}$$

where $\tilde{\mathbf{v}}_{w,j}$ is the $j^{th}$ maximal right singular vector of $\boldsymbol{\Lambda}^T \mathbf{X}$. However, notice that the layer weights in equation 30 are the optimal weights for the relaxed problem, i.e.,

$$\min_{\{\theta_l\}_{l=1}^L} \sum_{j=1}^{m_{L-1}} \|\mathbf{w}_{L,j}\|_2 \text{ s.t. } \sum_{j=1}^{m_{L-1}} \mathbf{X} \mathbf{W}_1^j \ldots \mathbf{w}_{L-1}^j \mathbf{w}_{L,j}^T = \mathbf{Y}, \forall \theta_l^j \in \Theta_{L-1}. \tag{31}$$

Using the optimal layer weights in equation 30, we have the following network output for the relaxed model

$$\sum_{j=1}^{m_{L-1}} \mathbf{X}\mathbf{W}_1^j \dots \mathbf{w}_{L-1}^j \mathbf{w}_{L,j}^T = \gamma \sum_{j=1}^{m_{L-1}} \mathbf{q}_{w,j} \mathbf{w}_{L,j}^T.$$

Since we know that the objective value for equation 31 is a lower bound for equation 15, the layer weights that achieve the output above for the original problem in equation 15 is optimal. Thus, a set of optimal solutions to equation 15 can be formulated as follows

$$\mathbf{W}_l^* = \begin{cases} t^* \sum_{j=1}^{m_{L-1}} \tilde{\mathbf{v}}_{w,j} \boldsymbol{\rho}_{1,j}^T & \text{if } l = 1 \\ t^* \sum_{j=1}^{m_{L-1}} \boldsymbol{\rho}_{l-1,j} \boldsymbol{\rho}_{l,j}^T & \text{if } 1 < l \le L - 2 \\ \sum_{j=1}^{m_{L-1}} \boldsymbol{\rho}_{L-2,j} & \text{if } l = L - 1 \end{cases} \tag{32}$$

where we select a set of unit norm vectors $\{\boldsymbol{\rho}_{l,j}\}_{l=1}^{L-2}$ such that $\boldsymbol{\rho}_{l,j}^T \boldsymbol{\rho}_{l,k} = 0, \ \forall j \ne k$. $\qquad\square$

**Theorem 3.2.** *Let $\{\mathbf{X}, \mathbf{y}\}$ be feasible for equation 14, then strong duality holds for finite width networks.*

***Proof of Theorem 3.2***. We first select a set of unit norm vectors, i.e., $\{\boldsymbol{\rho}_l\}_{l=1}^{L-2}$, to construct weight matrices $\{\mathbf{W}_l^e\}_{l=1}^{L-1}$ that satisfies equation 28. Then, the dual of equation 14 can be written as

$$D_e^* = \max_{\boldsymbol{\lambda}} \boldsymbol{\lambda}^T \mathbf{y}$$
$$\text{s.t. } |(\mathbf{X}\mathbf{W}_1^e \dots \mathbf{w}_{L-1}^e)^T \boldsymbol{\lambda}| \le 1$$

Then, we have

$$P^* = \min_{\{\theta_l\}_{l=1}^{L-1} \in \Theta_{L-1}} \max_{\boldsymbol{\lambda}} \boldsymbol{\lambda}^T \mathbf{y} \qquad \ge \qquad \max_{\boldsymbol{\lambda}} \boldsymbol{\lambda}^T \mathbf{y} \tag{33}$$
$$\text{s.t. } |(\mathbf{X}\mathbf{W}_1 \dots \mathbf{w}_{L-1})^T \boldsymbol{\lambda}| \le 1 \qquad \text{s.t. } |(\mathbf{X}\mathbf{W}_1 \dots \mathbf{w}_{L-1})^T \boldsymbol{\lambda}| \le 1, \ \forall \theta_l \in \Theta_{L-1}$$
$$= \max_{\boldsymbol{\lambda}} \boldsymbol{\lambda}^T \mathbf{y}$$
$$\text{s.t. } |(\mathbf{X}\mathbf{W}_1^e \dots \mathbf{w}_{L-1}^e)^T \boldsymbol{\lambda}| \le 1$$
$$= D_e^* = D^* = D_m^*$$

where the first inequality follows from changing the order of min-max to obtain a lower bound and the first equality follows from the fact that $\{\mathbf{W}_l^e\}_{l=1}^{L-1}$ maximizes the dual problem. Furthermore, we have the following relation between the primal problems

$$P_e^* = \min_{\mathbf{w}_L} \|\mathbf{w}_L\|_1 \qquad \ge \qquad P^* = \min_{\{\theta_l\}_{l=1}^{L} \in \Theta_{L-1}} \|\mathbf{w}_L\|_1 \tag{34}$$
$$\text{s.t. } \mathbf{W}_1^e \dots \mathbf{W}_{L-1}^e \mathbf{w}_L = \mathbf{y} \qquad \text{s.t. } \mathbf{W}_1 \dots \mathbf{W}_{L-1} \mathbf{w}_L = \mathbf{y},$$

where the inequality follows from the fact that the original problem has infinite width in each layer. Now, notice that the optimization problem on the left hand side of equation 34 is convex since it is an $\ell_1$-norm minimization problem with linear equality constraints. Therefore, strong duality holds for this problem, i.e., $P_e^* = D_e^*$ and we have $P_e^* \ge P^* \ge P_m^* \ge D_e^* = D^* = D_m^*$. Using this result along with equation 33, we prove that strong duality holds, i.e., $P_e^* = P^* = P_m^* = D_e^* = D^* = D_m^*$.

$\qquad\square$

**Corollary 3.1.** *Theorem 3.1 implies that deep linear networks can obtain a scaled version of $\mathbf{y}$ using only the first layer, i.e., $\mathbf{X}\mathbf{W}_1 \boldsymbol{\rho}_1 = c\mathbf{y}$, where $c > 0$. Therefore, the remaining layers do not contribute to the expressive power.*

***Proof of Corollary 3.1***. The proof directly follows from equation 28. $\qquad\square$

**Corollary A.1.** *The analysis above and Theorem 3.2 also show that strong duality holds for the regularized deep linear network training problem.*

***Proof of Corollary A.1.*** The proof directly follows from the analysis in this section and Theorem 3.2. □

**Lemma A.4.** *The following problems are equivalent:*

$$\min_{\{\theta_l\}_{l=1}^{L}} \sum_{l=1}^{L} \|\mathbf{W}_l\|_F^2 \quad = \quad \min_{\{\theta_l\}_{l=1}^{L}, \{t_l\}_{l=1}^{L-2}} \sum_{j=1}^{m_{L-1}} \|\mathbf{w}_{L,j}\|_2 + \sum_{l=1}^{L-2} t_l^2 .$$
$$\text{s.t. } f_{\theta,L}(\mathbf{X}) = \mathbf{Y} \qquad \text{s.t. } f_{\theta,L}(\mathbf{X}) = \mathbf{Y}, \ \mathbf{w}_{L-1,j} \in \mathcal{B}_2, \|\mathbf{W}_l\|_F \le t_l, \ \forall l \in [L-2]$$

***Proof of Lemma A.4.*** Applying the scaling trick in Lemma A.1 to the last two layer of the $L$-layer network in equation 15 gives

$$\min_{\{\theta_l\}_{l=1}^{L}, \{t_l\}_{l=1}^{L-2}} \sum_{j=1}^{m_{L-1}} \|\mathbf{w}_{L,j}\|_2 + \sum_{l=1}^{L-2} \|\mathbf{W}_l\|_F^2 .$$
$$\text{s.t. } \|\mathbf{w}_{L-1,j}\|_2 \le 1, \forall j \in [m_{L-1}]$$
$$\mathbf{X}\mathbf{W}_1 \dots \mathbf{W}_{L-1}\mathbf{W}_L = \mathbf{Y}$$

Then, we use the epigraph form for the norm of the first $L-2$ to achieve the equivalence. □

**Theorem 3.4.** *Let $\{\mathbf{X}, \mathbf{y}\}$ be feasible for equation 15, then strong duality holds for finite width networks.*

***Proof of Theorem 3.4.*** We first select a set of unit norm vectors, i.e., $\{\boldsymbol{\rho}_{l,j}\}_{l=1}^{L-2}$, to construct weight matrices $\{\mathbf{W}_l^{e,j}\}_{l=1}^{L-1}$ that satisfies equation 30. Then, the dual of equation 15 can be written as

$$D_e^* = \max_{\boldsymbol{\Lambda}} \text{trace}(\boldsymbol{\Lambda}^T \mathbf{Y})$$
$$\text{s.t. } \sigma_{max}(\boldsymbol{\Lambda}^T \mathbf{X}\mathbf{W}_1^{e,j} \dots \mathbf{W}_{L-2}^{e,j}) \le 1, \ \forall j$$

Then, we have

$$P^* = \min_{\{\theta_l\}_{l=1}^{L-1} \in \Theta_{L-1}} \max_{\boldsymbol{\Lambda}} \text{trace}(\boldsymbol{\Lambda}^T \mathbf{Y}) \qquad \ge \quad \max_{\boldsymbol{\Lambda}} \text{trace}(\boldsymbol{\Lambda}^T \mathbf{Y}) \qquad (35)$$
$$\text{s.t. } \sigma_{max}(\boldsymbol{\Lambda}^T \mathbf{X}\mathbf{W}_1 \dots \mathbf{W}_{L-2}) \le 1 \qquad \text{s.t. } \sigma_{max}(\boldsymbol{\Lambda}^T \mathbf{X}\mathbf{W}_1 \dots \mathbf{W}_{L-2}) \le 1, \ \forall \theta_l \in \Theta_{L-1}$$
$$= \quad \max_{\boldsymbol{\Lambda}} \text{trace}(\boldsymbol{\Lambda}^T \mathbf{Y})$$
$$\text{s.t. } \sigma_{max}(\boldsymbol{\Lambda}^T \mathbf{X}\mathbf{W}_1^{e,j} \dots \mathbf{W}_{L-2}^{e,j}) \le 1, \ \forall j$$
$$= \quad D_e^* = D^* = D_m^*$$

where the first inequality follows from changing the order of min-max to obtain a lower bound and the first equality follows from the fact that $\{\mathbf{W}_l^{e,j}\}_{l=1}^{L-1}$ maximizes the dual problem. Furthermore, we have the following relation between the primal problems

$$P_e^* = \min_{\mathbf{W}_L} \sum_{j=1}^{m_{L-1}} \|\mathbf{w}_{L,j}\|_2 \qquad \ge \qquad P^* = \min_{\{\theta_l\}_{l=1}^{L} \in \Theta_{L-1}} \sum_{j=1}^{m_{L-1}} \|\mathbf{w}_{L,j}\|_2 \quad (36)$$
$$\text{s.t. } \sum_{j=1}^{m_{L-1}} \mathbf{W}_1^{e,j} \dots \mathbf{W}_{L-1}^{e,j} \mathbf{w}_{L,j}^T = \mathbf{Y} \qquad \text{s.t. } \mathbf{W}_1 \dots \mathbf{W}_{L-1}\mathbf{W}_L = \mathbf{Y},$$

where the inequality follows from the fact that the original problem has infinite width in each layer. Now, notice that the optimization problem on the left hand side of equation 36 is convex since it is an $\ell_2$-norm minimization problem with linear equality constraints. Therefore, strong duality holds for this problem, i.e., $P_e^* = D_e^*$ and we have $P_e^* \ge P^* \ge P_m^* \ge D_e^* = D^* = D_m^*$. Using this result along with equation 35, we prove that strong duality holds, i.e., $P_e^* = P^* = P_m^* = D_e^* = D^* = D_m^*$.

□

A.7   PROOFS FOR THE DEEP RELU NETWORKS

**Theorem 4.1.** *Let $\mathbf{X}$ be a rank-one data matrix such that $\mathbf{X} = \mathbf{c}\mathbf{a}_0^T$, where $\mathbf{c} \in \mathbb{R}_+^n$ and $\mathbf{a}_0 \in \mathbb{R}^d$, then strong duality holds and the optimal weights for each layer can be formulated as follows*

$$\mathbf{W}_l = \frac{\boldsymbol{\phi}_{l-1}}{\|\boldsymbol{\phi}_{l-1}\|_2}\boldsymbol{\phi}_l^T, \ \forall l \in [L-2], \ \mathbf{w}_{L-1} = \frac{\boldsymbol{\phi}_{L-2}}{\|\boldsymbol{\phi}_{L-2}\|_2},$$

*where $\boldsymbol{\phi}_0 = \mathbf{a}_0$ and $\{\boldsymbol{\phi}_l\}_{l=1}^{L-2}$ is a set of vectors such that $\boldsymbol{\phi}_l \in \mathbb{R}_+^{m_l}$ and $\|\boldsymbol{\phi}_l\|_2 = t^*, \ \forall l \in [L-2]$.*

**Proposition 1.** *First $L-2$ hidden layer weight matrices in equation 16 have the same operator and Frobenius norms.*

***Proof of Proposition 1***. Let us first denote the sum of the norms for the first $L-2$ layer as $t$, i.e., $t = \sum_{l=1}^{L-2} t_l$, where $t_l = \|\mathbf{W}_l\|_2 = \|\mathbf{W}_l\|_F$ since the upper-bound is achieved when the matrices are rank-one. Then, to find the extreme points (see the details in Proof of Theorem 4.1), we need to solve the following problem

$$\underset{\{\theta_l\}_{l=1}^{L-2}}{\arg\max} |\boldsymbol{\lambda}^{*T}\mathbf{c}| \|\mathbf{a}_{L-2}\|_2 = \underset{\{\theta_l\}_{l=1}^{L-2} \in \Theta_{L-1}}{\arg\max} |\boldsymbol{\lambda}^{*T}\mathbf{c}| \|(\mathbf{a}_{L-3}^T\mathbf{W}_{L-2})_+\|_2$$

where we use $\mathbf{a}_{L-2}^T = (\mathbf{a}_{L-3}^T\mathbf{W}_{L-2})_+$. Since $\|\mathbf{W}_{L-2}\|_F = t_{L-2} = t - \sum_{l=1}^{L-3}$, the objective value above becomes $|\boldsymbol{\lambda}^{*T}\mathbf{c}| \|(\mathbf{a}_{L-3}\|_2 \left(t - \sum_{l=1}^{L-3}\right)$. Applying this step to all the remaining layer weights gives the following problem

$$\underset{\{t_l\}_{l=1}^{L-3} \in \Theta_{L-1}}{\arg\max} |\boldsymbol{\lambda}^{*T}\mathbf{c}| \|\mathbf{a}_0\|_2 \left(t - \sum_{l=1}^{L-3}\right)\prod_{j=1}^{L-3} t_l \text{ s.t. s.t. } \sum_{l=1}^{L-3} t_l \leq t, \ t_l \geq 0.$$

Then, the proof directly follows from Proof of Proposition 3.1. □

***Proof of Theorem 4.1***. Using Lemma A.3 and Proposition 1, this problem can be equivalently stated as

$$\underset{\{\theta_l\}_{l=1}^{L} \in \Theta_{L-1}}{\min} \|\mathbf{w}_L\|_1 \text{ s.t. } \mathbf{A}_l = (\mathbf{A}_{l-1}\mathbf{W}_l)_+, \ \forall l \in [L-1]$$
$$\mathbf{A}_{L-1}\mathbf{w}_L = \mathbf{y}$$
(37)

which also has the following dual form

$$P^* = \underset{\{\theta_l\}_{l=1}^{L-1} \in \Theta_{L-1}}{\min} \underset{\boldsymbol{\lambda}}{\max} \boldsymbol{\lambda}^T\mathbf{y}$$
$$\text{s.t. } \|\mathbf{A}_{L-1}^T\boldsymbol{\lambda}\|_\infty \leq 1$$
(38)

Notice that we remove the recursive constraint in equation 38 for notational simplicity, however, $\mathbf{A}_{L-1}$ is still a function of all the layer weights except $\mathbf{w}_L$. Changing the order of min-max in equation 38 gives

$$P^* \geq D^* = \underset{\boldsymbol{\lambda}}{\max} \boldsymbol{\lambda}^T\mathbf{y} \text{ s.t. } \|\mathbf{A}_{L-1}^T\boldsymbol{\lambda}\|_\infty \leq 1, \ \forall \theta_l \in \Theta_{L-1}, \ \forall l \in [L-1].$$
(39)

The dual of the semi-infinite problem in equation 39 is given by

$$\min \|\boldsymbol{\mu}\|_{TV}$$
$$\text{s.t. } \int_{\{\theta_l\}_{l=1}^{L-1} \in \Theta_{L-1}} (\mathbf{A}_{L-2}\mathbf{w}_{L-1})_+ d\boldsymbol{\mu}(\theta_1, \ldots, \theta_{L-1}) = \mathbf{y},$$
(40)

where $\boldsymbol{\mu}$ is a signed Radon measure and $\|\cdot\|_{TV}$ is the total variation norm. We emphasize that equation 40 has infinite width in each layer, however, an application of Caratheodory's theorem shows that the measure $\boldsymbol{\mu}$ in the integral can be represented by finitely many (at most $n+1$) Dirac delta functions (Rosset et al., 2007). Thus, we choose

$$\boldsymbol{\mu} = \sum_{j=1}^{m_{L-1}} \delta(\mathbf{W}_1 - \mathbf{W}_1^j, \ldots, \mathbf{w}_{L-1} - \mathbf{w}_{L-1}^j)w_{L,j},$$

where $\delta(\cdot)$ is the Dirac delta function and the superscript indicates a particular choice for the corresponding layer weight. This selection of $\boldsymbol{\mu}$ yields the following problem

$$P_m^* = \min_{\{\theta_l\}_{l=1}^L} \|\mathbf{w}_L\|_1$$
$$\text{s.t.} \sum_{j=1}^{m_{L-1}} \left(\mathbf{A}_{L-2}^j \mathbf{w}_{L-1}^j\right)_+ w_{L,j} = \mathbf{y}, \ \theta_l^j \in \Theta_{L-1}, \ \forall l \in [L-1] \tag{41}$$

Here, we first note that even though the model in equation 41 has the same layer widths with regular deep ReLU networks, it has more expressive power since it allows us to choose multiple weight matrices for each layer. Based on this observation, we have $P^* \geq P_m^*$.

As a consequence of equation 39, we can characterize the optimal layer weights for equation 41 as the extreme points that solve

$$\arg\max_{\{\theta_l\}_{l=1}^{L-1} \in \Theta_{L-1}} |\boldsymbol{\lambda}^{*T} (\mathbf{A}_{L-2}\mathbf{w}_{L-1})_+| \tag{42}$$

where $\boldsymbol{\lambda}^*$ is the optimal dual parameter. Since we assume that $\mathbf{X} = \mathbf{ca}_0^T$ with $\mathbf{c} \in \mathbb{R}_+^n$, we have $\mathbf{A}_{L-2} = \mathbf{ca}_{L-2}^T$, where $\mathbf{a}_l^T = (\mathbf{a}_{l-1}^T \mathbf{W}_l)_+$, $\mathbf{a}_l \in \mathbb{R}_+^{m_l}$ and $\forall l \in [L-1]$. Based on this observation, we have $\mathbf{w}_{L-1} = \mathbf{a}_{L-2}/\|\mathbf{a}_{L-2}\|_2$, which reduces equation 42 to the following

$$\arg\max_{\{\theta_l\}_{l=1}^{L-2} \in \Theta_{L-1}} |\boldsymbol{\lambda}^{*T}\mathbf{c}| \, \|\mathbf{a}_{L-2}\|_2 \tag{43}$$

We then apply the same approach to all the remaining layer weights. However, notice that each neuron for the first $L-2$ layers must have bounded Frobenius norms due to the norm constraint. If we denote the optimal $\ell_2$ norms vector for the neuron in the $l^{\text{th}}$ layer as $\phi_l \in \mathbb{R}_+^{m_l}$, then we have the following formulation for the layer weights that solve equation 42

$$\mathbf{W}_l = \frac{\phi_{l-1}}{\|\phi_{l-1}\|_2} \phi_l^T, \ \forall l \in [L-2], \ \mathbf{w}_{L-1} = \frac{\phi_{L-2}}{\|\phi_{L-2}\|_2}, \tag{44}$$

where $\phi_0 = \mathbf{a}_0$, $\{\phi_l\}_{l=1}^{L-2}$ is a set of nonnegative vectors satisfying $\|\phi_l\|_2 = t^*$, $\forall l \in [L-2]$.

We note that the layer weights in equation 44 are optimal for the relaxed problem in equation 41. However, since there exists a single possible choice for the left singular vector of $\mathbf{W}_1$ and we can select an arbitrary set for $\{\phi_l\}_{l=1}^{L-2}$, the dual problems coincide for equation 16 and equation 41, i.e., we achieve $D_m^* = D^*$ using the same layer weights, where $D_m^*$ is the optimal dual objective value for equation 41. Therefore, the set of weights in equation 44 are also optimal for equation 16. $\qquad\square$

**Theorem 4.2.** *Let $\mathbf{X}$ be a data matrix such that $\mathbf{X} = \mathbf{ca}_0^T$, where $\mathbf{c} \in \mathbb{R}^n$ and $\mathbf{a}_0 \in \mathbb{R}^d$. Then, a set of optimal solutions to equation 16 satisfies $\{(\mathbf{w}_i, b_i)\}_{i=1}^m$, where $\mathbf{w}_i = s_i \frac{\mathbf{a}_0}{\|\mathbf{a}_0\|_2}, b_i = -s_i c_i \|\mathbf{a}_0\|_2$ with $s_i = \pm 1, \forall i \in [m]$.*

***Proof of Theorem 4.2.*** Given $\mathbf{X} = \mathbf{ca}_0^T$, all possible extreme points can be characterized as follows

$$\arg\max_{b,\mathbf{w}:\|\mathbf{w}\|_2=1} |\boldsymbol{\lambda}^T(\mathbf{Xw} + b\mathbf{1})_+| = \arg\max_{b,\mathbf{w}:\|\mathbf{w}\|_2=1} |\boldsymbol{\lambda}^T(\mathbf{ca}_0^T\mathbf{w} + b\mathbf{1})_+|$$
$$= \arg\max_{b,\mathbf{w}:\|\mathbf{w}\|_2=1} \left| \sum_{i=1}^n \lambda_i (c_i \mathbf{a}_0^T\mathbf{w} + b)_+ \right|$$

which can be equivalently stated as

$$\arg\max_{b,\mathbf{w}:\|\mathbf{w}\|_2=1} \sum_{i\in\mathcal{S}} \lambda_i c_i \mathbf{a}_0^T\mathbf{w} + \sum_{i\in\mathcal{S}} \lambda_i b \ \text{s.t.} \ \begin{cases} c_i \mathbf{a}_0^T\mathbf{w} + b \geq 0, \forall i \in \mathcal{S} \\ c_j \mathbf{a}_0^T\mathbf{w} + b \leq 0, \forall j \in \mathcal{S}^c \end{cases},$$

which shows that $\mathbf{w}$ must be either positively or negatively aligned with $\mathbf{a}_0$, i.e., $\mathbf{w} = s\frac{\mathbf{a}_0}{\|\mathbf{a}_0\|_2}$, where $s = \pm 1$. Thus, $b$ must be in the range of $[\max_{i\in\mathcal{S}}(-sc_i\|\mathbf{a}_0\|_2), \ \min_{j\in\mathcal{S}^c}(-sc_j\|\mathbf{a}_0\|_2)]$ Using these observations, extreme points can be formulated as follows

$$\mathbf{w}_\lambda = \begin{cases} \frac{\mathbf{a}_0}{\|\mathbf{a}_0\|_2} & \text{if } \sum_{i\in\mathcal{S}} \lambda_i c_i \geq 0 \\ \frac{-\mathbf{a}_0}{\|\mathbf{a}_0\|_2} & \text{otherwise} \end{cases} \ \text{and } b_\lambda = \begin{cases} \min_{j\in\mathcal{S}^c}(-s_\lambda c_j \|\mathbf{a}_0\|_2) & \text{if } \sum_{i\in\mathcal{S}} \lambda_i \geq 0 \\ \max_{i\in\mathcal{S}}(-s_\lambda c_i \|\mathbf{a}_0\|_2) & \text{otherwise} \end{cases},$$

where $s_\lambda = \text{sign}(\sum_{i\in\mathcal{S}} \lambda_i c_i)$. $\qquad\square$

**Proposition 4.1.** *Theorem 4.1 still holds when we add a bias term to the last hidden layer, i.e., the output becomes $\left(\mathbf{A}_{L-2}\mathbf{W}_{L-1} + \mathbf{1}_n\mathbf{b}^T\right)_+ \mathbf{w}_L = \mathbf{y}$, where $\mathbf{A}_l = (\mathbf{A}_{l-1}\mathbf{W}_l)_+, \ \forall l \in [L-2]$.*

***Proof of Proposition 4.1.*** Here, we add biases to the neurons in the last hidden layer of equation 16. For this case, all the equations in equation 37-equation 39 hold except notational changes due to the bias term. Thus, equation 42 changes as

$$
\underset{\{\theta_l\}_{l=1}^{L-1}\in\Theta_{L-1},b}{\arg\max}|\boldsymbol{\lambda}^{*^T}(\mathbf{A}_{L-2}\mathbf{w}_{L-1} + b\mathbf{1}_n)_+| = \underset{\{\theta_l\}_{l=1}^{L-1}\in\Theta_{L-1},b}{\arg\max}|\boldsymbol{\lambda}^{*^T}\left(\mathbf{ca}_{L-2}^T\mathbf{w}_{L-1} + b\mathbf{1}_n\right)_+|
$$

$$
= \underset{\{\theta_l\}_{l=1}^{L-2}\in\Theta_{L-1},b}{\arg\max}\left|\sum_{i=1}^n \lambda_i^*\left(c_i\mathbf{a}_{L-1}^T\mathbf{w}_{L-1} + b\right)_+\right|
$$

(45)

which can also be written as

$$
\underset{\{\theta_l\}_{l=1}^{L-1}\in\Theta_{L-1},b}{\arg\max}\sum_{i\in\mathcal{S}}\lambda_i^*c_i\mathbf{a}_{L-2}^T\mathbf{w}_{L-1} + \sum_{i\in\mathcal{S}}\lambda_i^*b \ \text{s.t.} \ \begin{cases} c_i\mathbf{a}_{L-2}^T\mathbf{w}_{L-1} + b \geq 0, \forall i \in \mathcal{S} \\ c_j\mathbf{a}_{L-2}^T\mathbf{w}_{L-1} + b \leq 0, \forall j \in \mathcal{S}^c \end{cases},
$$

where $\mathcal{S}$ and $\mathcal{S}^c$ are the indices for which ReLU is active and inactive, respectively. This shows that $\mathbf{w}_{L-1}$ must be $\mathbf{w}_{L-1} = \pm 1\frac{\mathbf{a}_{L-2}}{\|\mathbf{a}_{L-2}\|_2}$ and $b \in [\max_{i\in\mathcal{S}}(-c_i\|\mathbf{a}_{L-2}\|_2), \ \min_{j\in\mathcal{S}^c}(-c_j\|\mathbf{a}_{L-2}\|_2)]$. Then, we obtain the following

$$
\mathbf{w}_{L-1}^* = \begin{cases} \frac{\mathbf{a}_{L-2}}{\|\mathbf{a}_{L-2}\|_2} & \text{if } \sum_{i\in\mathcal{S}}\lambda_i^*c_i \geq 0 \\ \frac{-\mathbf{a}_{L-2}}{\|\mathbf{a}_{L-2}\|_2} & \text{otherwise} \end{cases} \text{ and } b^* = \begin{cases} \min_{j\in\mathcal{S}^c}(-s_{\lambda^*}c_j\|\mathbf{a}_{L-2}\|_2) & \text{if } \sum_{i\in\mathcal{S}}\lambda_i^* \geq 0 \\ \max_{i\in\mathcal{S}}(-s_{\lambda^*}c_i\|\mathbf{a}_{L-2}\|_2) & \text{otherwise} \end{cases},
$$

(46)

where $s_{\lambda^*} = \text{sign}(\sum_{i\in\mathcal{S}}\lambda_i^*c_i)$. This result reduces equation 45 to the following problem

$$
\underset{\{\theta_l\}_1^{L-2}\in\Theta_{L-1}}{\arg\max}|C(\boldsymbol{\lambda}^*, \mathbf{c})|\|\mathbf{a}_{L-2}\|_2,
$$

where $C(\boldsymbol{\lambda}^*, \mathbf{c})$ is constant scalar independent of $\{\mathbf{W}_l\}_{l=1}^{L-2}$. Hence, this problem and its solutions are the same with equation 43 and equation 44, respectively.

$\square$

**Corollary 4.1.** *As a result of Theorem 4.2, when we have one dimensional data, i.e., $\mathbf{x} \in \mathbb{R}^n$, an optimal solution to equation 16 can be formulated as $\{(w_i, b_i)\}_{i=1}^m$, where $\mathbf{w}_i = s_i$, $b_i = -s_ix_i$ with $s_i = \pm 1, \forall i \in [m]$. Therefore, the optimal network output has kinks only at the input data points, i.e., the output function is in the following form: $f_{\theta,2}(\hat{x}) = \sum_i (\hat{x} - x_i)_+$. Therefore, the network output becomes linear spline interpolation for one dimensional datasets.*

**Corollary 4.2.** *As a result of Theorem 4.2 and Proposition 4.1, when we have one dimensional data, i.e., $\mathbf{x} \in \mathbb{R}^n$, the optimal network output has kinks only at the input data points, i.e., the output function is in the following form: $f_{\theta,L}(\hat{x}) = \sum_i (\hat{x} - x_i)_+$. Therefore, the network output becomes linear spline interpolation for one dimensional datasets.*

***Proof of Corollary 4.1 and 4.2.*** Let us particularly consider the input sample $\mathbf{a}_0$. Then, the activations of the network defined by equation 44 and equation 46 are

$$
\mathbf{a}_1^T = (\mathbf{a}_0^T\mathbf{W}_1)_+ = \left(\mathbf{a}_0^T\frac{\mathbf{a}_0}{\|\mathbf{a}_0\|_2}\boldsymbol{\phi}_1^T\right)_+ = \|\mathbf{a}_0\|_2\boldsymbol{\phi}_1^T
$$

$$
\mathbf{a}_2^T = (\mathbf{a}_1^T\mathbf{W}_2)_+ = \left(\mathbf{a}_1^T\frac{\mathbf{a}_1}{\|\mathbf{a}_1\|_2}\boldsymbol{\phi}_2^T\right)_+ = \|\mathbf{a}_0\|_2\|\boldsymbol{\phi}_1^T\|_2\boldsymbol{\phi}_2^T
$$

$$
\vdots
$$

$$
\mathbf{a}_{L-2}^T = (\mathbf{a}_{L-3}^T\mathbf{W}_{L-2})_+ = \left(\mathbf{a}_{L-3}^T\frac{\mathbf{a}_{L-3}}{\|\mathbf{a}_{L-3}\|_2}\boldsymbol{\phi}_{L-2}^T\right)_+ = \|\mathbf{a}_0\|_2\|\boldsymbol{\phi}_1^T\|_2\ldots\|\boldsymbol{\phi}_{L-3}^T\|_2\boldsymbol{\phi}_{L-2}^T
$$

$$
a_{L-1} = (\mathbf{a}_{L-2}^T\mathbf{w}_{L-1} + b)_+ = (\|\mathbf{a}_{L-2}\|_2 - \|\mathbf{a}_{L-2}\|_2)_+ = 0.
$$

Thus, if we feed $c_i\mathbf{a}_0$ to the network, we get $a_{L-1} = (c_i\|\mathbf{a}_{L-2}\|_2 - c_i\|\mathbf{a}_{L-2}\|_2)_+ = 0$, where we use the fact that optimal biases are in the form of $b = -c_i\|\mathbf{a}_{L-2}\|_2$ as proved in equation 46. This analysis proves that the kink of each ReLU activation occurs exactly at one of the data points. $\square$

**Proposition 4.2.** *Strong duality also holds for deep ReLU networks with vector outputs and the optimal layer weights can be formulated as in Theorem 4.1.*

***Proof of Proposition 4.2.*** For vector outputs, we have the following training problem

$$\min_{\{\theta_l\}_{l=1}^L} \sum_{l=1}^L \|\mathbf{W}_l\|_F^2 \text{ s.t. } f_{\theta,L}(\mathbf{X}) = \mathbf{Y}.$$

After a suitable rescaling as in the previous case, the above problem has the following dual

$$P^* \geq D^* = \max_{\boldsymbol{\lambda}} \text{trace}(\boldsymbol{\Lambda}^T \mathbf{Y}) \text{ s.t. } \|\boldsymbol{\Lambda}^T (\mathbf{A}_{L-2}\mathbf{w}_{L-1})_+\|_2 \leq 1, \ \forall \theta_l \in \Theta_{L-1}, \ \forall l \in [L-1]. \quad (47)$$

Using equation 47, we can characterize the optimal layer weights as the extreme points that solve

$$\underset{\{\theta_l\}_{l=1}^{L-1} \in \Theta_{L-1}}{\arg\max} \ \|\boldsymbol{\Lambda}^{*T}(\mathbf{A}_{L-2}\mathbf{w}_{L-1})_+\|_2, \quad (48)$$

where $\boldsymbol{\Lambda}^*$ is the optimal dual parameter. Since we assume that $\mathbf{X} = \mathbf{c}\mathbf{a}_0^T$ with $\mathbf{c} \in \mathbb{R}_+^n$, we have $\mathbf{A}_{L-2} = \mathbf{c}\mathbf{a}_{L-2}^T$, where $\mathbf{a}_l^T = (\mathbf{a}_{l-1}^T\mathbf{W}_l)_+$, $\mathbf{a}_l \in \mathbb{R}_+^{m_l}$ and $\forall l \in [L-1]$. Based on this observation, we have $\mathbf{w}_{L-1} = \mathbf{a}_{L-2}/\|\mathbf{a}_{L-2}\|_2$, which reduces equation 48 to the following

$$\underset{\{\theta_l\}_{l=1}^{L-2} \in \Theta_{L-1}}{\arg\max} \ \|\boldsymbol{\Lambda}^{*T}\mathbf{c}\|_2 \|\mathbf{a}_{L-2}\|_2.$$

Then, the rest of steps directly follow Theorem 4.1 yielding the following weight matrices

$$\mathbf{W}_l = \frac{\boldsymbol{\phi}_{l-1}}{\|\boldsymbol{\phi}_{l-1}\|_2}\boldsymbol{\phi}_l^T, \ \forall l \in [L-2], \ \mathbf{w}_{L-1} = \frac{\boldsymbol{\phi}_{L-2}}{\|\boldsymbol{\phi}_{L-2}\|_2},$$

where $\boldsymbol{\phi}_0 = \mathbf{a}_0$, $\{\boldsymbol{\phi}_l\}_{l=1}^{L-2}$ is a set of nonnegative vectors satisfying $\|\boldsymbol{\phi}_l\|_2 = t^*$, $\forall l \in [L-2]$. Moreover, as a direct consequence of Theorem 3.4, strong duality holds for deep ReLU networks. □

**Theorem 4.3.** *Let $\{\mathbf{X}, \mathbf{Y}\}$ be a dataset such that $\mathbf{X}\mathbf{X}^T = \mathbf{I}_n$[11] and $\mathbf{Y}$ has orthogonal columns, then the optimal weight matrices for each layer can be formulated as follows*

$$\mathbf{W}_l = \frac{1}{\sqrt{2K}} \sum_{r=1}^{2K} \frac{\boldsymbol{\phi}_{l-1,r}}{\|\boldsymbol{\phi}_{l-1,r}\|_2}\boldsymbol{\phi}_{l,r}^T, \ \forall l \in [L-2], \ \mathbf{W}_{L-1} = \frac{1}{\sqrt{2K}} \left[ \frac{\boldsymbol{\phi}_{L-2,1}}{\|\boldsymbol{\phi}_{L-2,1}\|_2} \quad \cdots \quad \frac{\boldsymbol{\phi}_{L-2,2K}}{\|\boldsymbol{\phi}_{L-2,2K}\|_2} \right],$$

*where $(\boldsymbol{\phi}_{0,2j-1}, \boldsymbol{\phi}_{0,2j}) = \left(\mathbf{X}^T(\mathbf{y}_j)_+, \mathbf{X}^T(-\mathbf{y}_j)_+\right)$, $\forall j \in [K]$ and $\{\boldsymbol{\phi}_{l,r}\}_{l=1}^{L-2}$ is a set of vectors such that $\boldsymbol{\phi}_{l,r} \in \mathbb{R}_+^{m_l}$, $\|\boldsymbol{\phi}_{l,r}\|_2 = t^*$, and $\boldsymbol{\phi}_{l,i}^T\boldsymbol{\phi}_{l,j} = 0$, $\forall i \neq j$.*

***Proof of Theorem 4.3.*** For vector outputs, we have the following training problem

$$\min_{\{\theta_l\}_{l=1}^L} \sum_{l=1}^L \|\mathbf{W}_l\|_F^2 \text{ s.t. } f_{\theta,L}(\mathbf{X}) = \mathbf{Y}.$$

After a suitable rescaling as in the previous case, the above problem has the following dual

$$P^* \geq D^* = \max_{\boldsymbol{\lambda}} \text{trace}(\boldsymbol{\Lambda}^T \mathbf{Y}) \text{ s.t. } \|\boldsymbol{\Lambda}^T (\mathbf{A}_{L-2}\mathbf{w}_{L-1})_+\|_2 \leq 1, \ \forall \theta_l \in \Theta_{L-1}, \ \forall l \in [L-1]. \quad (49)$$

Using equation 49, we can characterize the optimal layer weights as the extreme points that solve

$$\underset{\{\theta_l\}_{l=1}^{L-1} \in \Theta_{L-1}}{\arg\max} \ \|\boldsymbol{\Lambda}^{*T}(\mathbf{A}_{L-2}\mathbf{w}_{L-1})_+\|_2, \quad (50)$$

where $\boldsymbol{\Lambda}^*$ is the optimal dual parameter. We first note that since $\mathbf{X}$ is whitened such that $\mathbf{X}\mathbf{X}^T = \mathbf{I}_n$, equation 50 implies $\sigma_{max}(\boldsymbol{\Lambda}^*) \leq t^{*L-2}$. Then, the objective is trivially maximized by $\boldsymbol{\Lambda}^* =$

---

[11]This can be achieved by applying batch whitening, which often improves accuracy (Huang et al., 2018).

$t^{*L-2}\mathbf{Y}/\sigma_{max}(\mathbf{Y})$, which is also a feasible solution. Therefore, equation 50 can be equivalently written as

$$\underset{\{\theta_l\}_{l=1}^{L-1}\in\Theta_{L-1}}{\arg\max}\;\|\mathbf{Y}^T(\mathbf{A}_{L-2}\mathbf{w}_{L-1})_+\|_2. \tag{51}$$

We now note that since $\mathbf{Y}$ has orthogonal columns, equation 51 can be decomposed into $k$ maximization problems each of which can be maximized independently to find a set of extreme points. In particular, the $j^{th}$ problem can be formulated as follows

$$\underset{\{\theta_l\}_{l=1}^{L-1}\in\Theta_{L-1}}{\arg\max}\;|\mathbf{y}_j^T(\mathbf{A}_{L-2}\mathbf{w}_{L-1})_+| \le \max\left\{\|(\mathbf{y}_j)_+\|_2, \|(-\mathbf{y}_j)_+\|_2\right\}.$$

Then, noting the whitened data assumption, the rest of steps directly follow Theorem 4.1 yielding the following weight matrices

$$\mathbf{W}_l = \frac{1}{\sqrt{2}}\sum_{r=1}^{2}\frac{\phi_{l-1,r}}{\|\phi_{l-1,r}\|_2}\phi_{l,r}^T,\;\forall l\in[L-2],\;\mathbf{W}_{L-1} = \frac{1}{\sqrt{2}}\left[\frac{\phi_{L-2,1}}{\|\phi_{L-2,1}\|_2}\quad\frac{\phi_{L-2,2}}{\|\phi_{L-2,2}\|_2}\right],$$

where $\phi_{0,r} = \mathbf{X}^T(\pm\mathbf{y}_j)_+$ and $\{\phi_{l,r}\}_{l=1}^{L-2}$ is a set of nonnegative vectors satisfying $\|\phi_{l,r}\|_2 = t^*$ and $\phi_{l,1}^T\phi_{l,2} = 0,\;\forall l$. Thus, combining the extreme points for each $j$ yields

$$\mathbf{W}_l = \frac{1}{\sqrt{2K}}\sum_{r=1}^{2K}\frac{\phi_{l-1,r}}{\|\phi_{l-1,r}\|_2}\phi_{l,r}^T,\;\forall l\in[L-2],\;\mathbf{W}_{L-1} = \frac{1}{\sqrt{2K}}\left[\frac{\phi_{L-2,1}}{\|\phi_{L-2,1}\|_2}\quad\cdots\quad\frac{\phi_{L-2,2K}}{\|\phi_{L-2,2K}\|_2}\right],$$

where $(\phi_{0,2j-1}, \phi_{0,2j}) = \left(\mathbf{X}^T(\mathbf{y}_j)_+, \mathbf{X}^T(-\mathbf{y}_j)_+\right),\;\forall j\in[K]$ and $\{\phi_{l,r}\}_{l=1}^{L-2}$ is a set of nonnegative vectors satisfying $\|\phi_{l,r}\|_2 = t^*$ and $\phi_{l,i}^T\phi_{l,j} = 0\;\forall i\ne j$. Moreover, as a direct consequence of Theorem 3.4, strong duality holds for deep ReLU networks. $\square$

**Theorem 4.4.** *Let $\{\mathbf{X},\mathbf{Y}\}$ be a dataset such that $\mathbf{X}^T\mathbf{X} = \mathbf{I}_n$ and $\mathbf{Y}$ has orthogonal columns, then a set of optimal layer weight matrices for the following regularized training problem*

$$\min_{\theta\in\Theta}\frac{1}{2}\|f_{\theta,L}(\mathbf{X}) - \mathbf{Y}\|_F^2 + \frac{\beta}{2}\sum_{l=1}^{L}\|\mathbf{W}_l\|_F^2 \tag{17}$$

*can be formulated as follows*

$$\mathbf{W}_l = \begin{cases}\sum_{r=1}^{2K}\frac{\phi_{l-1,r}}{\|\phi_{l-1,r}\|_2}\phi_{l,r}^T, & \text{if } 1\le l\le L-1 \\ \sum_{r=1}^{2K}(\|\phi_{0,r}\|_2 - \beta)\phi_{l-1,r}\hat{\mathbf{e}}_r^T & \text{if } l = L\end{cases},$$

*where $\hat{\mathbf{e}}_{2j-1} = \hat{\mathbf{e}}_{2j} = \mathbf{e}_j,\;\forall j\in[K]$, $\mathbf{e}_j$ is the $j^{th}$ ordinary basis vector, and the other definitions follows from Theorem 4.3 except $t^* = 1$.*

***Proof of Theorem 4.4.*** From Section A.4.2 and the proof of Theorem 4.3, the dual problem has a closed-form solution as follows

$$\boldsymbol{\lambda}_j^* = \begin{cases}\beta\frac{\mathbf{y}_j}{\|\mathbf{y}_j\|_2} & \text{if } \beta\le\|\mathbf{y}_j\|_2 \\ \mathbf{y}_j & \text{otherwise}\end{cases},\quad\forall j\in[K]. \tag{52}$$

and the corresponding extreme points of the constraint in

$$\mathbf{W}_l = \left\{\sum_{r=1}^{2K}\frac{\phi_{l-1,r}}{\|\phi_{l-1,r}\|_2}\phi_{l,r}^T,\quad\text{if } 1\le l\le L-1\right., \tag{53}$$

where the definitions follows from Theorem 4.3.

We now note that given the hidden layer weight in equation 53, the primal problem in equation 17 is convex and differentiable with respect to the output layer weight $\mathbf{W}_L$. Thus, we can find the optimal output layer weights by simply taking derivative and equating it to zero. Applying these steps yields the following output layer weights

$$\mathbf{W}_L = \sum_{r=1}^{2K}(\|\phi_{0,r}\|_2 - \beta)\phi_{L-1,r}\hat{\mathbf{e}}_r^T,$$

where $\hat{\mathbf{e}}_{2j-1} = \hat{\mathbf{e}}_{2j} = \mathbf{e}_j,\;\forall j\in[K]$ and $\mathbf{e}_j$ is the $j^{th}$ ordinary basis vector. $\square$

