# OpenReview forum: "Revealing the Structure of Deep Neural Networks via Convex Duality"
_ICLR.cc/2021/Conference — Reject_

### Official Review · AnonReviewer2 · 2020-10-25
**Analytical formulae for optimal weights of trained regularized neural networks based on a convex-dual formulation**

**Rating:** 8
**Confidence:** 3

**Review:**

The authors consider training neural networks with a variety of losses and regularization (such as weight decay).  The authors introduce a novel convex-dual formulation which allows them to characterize optimal solutions as being extreme points of particular convex sets.    For multi-layer linear networks, the authors prove that the optimal weight matrices have rank equal to the number of outputs of the network, and whose singular vectors align with those of neighboring layers.  For ReLU nets in one-dimension, the authors prove that optimal solutions act as linear spline interpolators (the kinks between linear pieces occur at data points), and the authors prove closed form expressions for optimal weights at intermediate layers when input data is whitened.

The conclusions of this paper are strong and apply to a wide variety of neural networks.  The extension of linear spline interpolation results from 2-layers to more than 2-layers is a significant contribution.  The empirical comparisons of the presented analytical formulae for optimal weights to solutions via SGD are impressive (even though they are only performed on MNIST and CIFAR10 datasets).  The results of this paper themselves form a substantial contribution to the field and motivate clear followup work to test the performance of the provided formulae to larger and more realistic datasets, along with weakening the whitening assumptions.  Analytical expressions for weights of trained deep ReLU neural networks is a significant development, and the authors may want to provide more commentary on its significance (possibly with a small review of the most closely related works in this sense).  While I am not an expert in this subfield, there appear to be multiple significant contributions.

Minor comment:
Spelling error in footnote 3.

---

> ### Author Response · Authors · 2020-11-16
> **Responses to AnonReviewer2**
>
> We would like to thank the reviewer for the feedback and comments. As pointed out by the reviewer, the contributions in our paper can be extended to more realistic and large scale experiments. Hence, we have updated the paper to further emphasize the significance of theory for potential followup studies.

---

### Official Review · AnonReviewer3 · 2020-10-26
**Optimal solutions versus stationary points, saddle points, local minima, or gradient dynamics**

**Rating:** 3
**Confidence:** 5

**Review:**

The paper provides theoretical analysis and numerical experiments to characterize the structure of hidden layers and a set of optimal solutions. After warm-up with two-layer neural networks, the paper provides main theoretical results for deep linear networks and deep ReLU networks. Finally, numerical results are presented to verify theoretical analysis.

The paper cites the previous results in the literature of gradient dynamics of deep linear networks, and compare the contributions with this paper. However, the comparison is not done appropriately in the following sense. These previous results analyze the gradient dynamics of deep linear networks, whereas this paper analyses the set of optimal solutions. The set of optimal solutions is more well studied in the literature of loss landscapes of deep linear networks, where stationary points, saddle points as well as the set of optimal solutions are well studied.

What this paper proves would be interesting if these results apply for the characterization of stationary points, saddle points, local minima, or gradient dynamics, instead of the set of globally optimal solutions. For example, alignment of the layers through gradient dynamics is studied in the previous work https://arxiv.org/abs/1810.02032, which is interesting because it is implicitly done by gradient dynamics. However, in the present paper, the alignment of the layers is explicitly imposed by the constraints or the regularization on the norm of each layer’s weight. This is trivial since with this constraints, the norm is minimized with the alignment, of course. There is no need to use duality, which is superficial for this type of results and is unnecessary.

The results on deep ReLU networks are also trivial because of the unrealistic assumptions on the data, as I explained in the following. For Theorem 4.1., the author assumes that X = c a_0^T, where each entry of c is nonnegative. This assumption makes ReLU to be trivial by the following observation: for any vector v in R^d, (c v^T)_+ = (\diag(c) V)_+ where V = [v … v]^T, which is a n by d matrix. Because c is nonnegative, (\diag(c) V)_+ = \diag(c) (V)_+. Therefore, (c v^T)_+ = \diag(c) (V)_+ = \diag(c) V^+ where V^+ = [(v)_+ … (v)_+]^T. This means that the activation (on or off for ReLU nonlinearity) is exactly same for all data (over the dimension of n). This makes the results trivial for optimal solutions. Again, this setting would be interesting if the results are about gradient descent dynamics or loss surface, instead of optimal solutions. In that case, the activation is still the same for all data in this setting, but changes during time or vary in loss surface. For Theorem 4.2, the assumption on XX^T = I_n makes the result trivial by the following observation: the optimal solutions are simply the ones that memorize the data by using I_n. In the theorem, the optimal solution essentially memorizes the label at the first layer, as I expected. Again this might potentially be interesting if this is about stationary points, gradient descent or local minima, because this is nontrivial for those points. But, for the optimal solution with XX^T = I_n, this is trivial. This paper does not consider the case without XX^T = I_n: but, if it considers the case, if d >= n, then optimal solution is still trivial, because we can still memories the label at the first layer in a similar way. The assumption of XX^T = I_n is a strictly stronger version of the assumption of d >= n. Therefore, the results on deep ReLU networks are also trivial.

I would recommend the authors to carefully read the papers in the literature of implicit regularization versus explicit regularization, and in the literature of gradient dynamics and loss landscape of deep linear networks. Those papers are motivated to study the problem beyond the optimal solutions. Understanding optimal solutions is also interesting but not in the same or similar setting and the setting where optimal solutions are trivial. For example, characterizing optimal solutions would be interesting for deep ReLU networks if we cannot easily construct optimal solutions, which is the not case in the setting of this paper as I described above.

------------
UPDATE AFTER AUTHORS RESPONSE: The authors did not address my concerns. The author response and discussion in the paper for the whitening is wrong: this paper use the whitening to mean $\bf{X}\bf{X}^T=\bf{I}_n$ where n is the data size, but the whitening in machine learning means $\bf{X}^T \bf{X}=\bf{I}_d$ where d is input dimension. This is completely different: the former makes the problem trivial whereas the latter does not. The explanation with the mini-batch is also wrong: optimization problem and the optimal solution are defined for the full dataset, and not for the mini-batch. For the memorization, I am not talking about label memorization, but the fact that we can set the weight matrix to have the direction of Y at the first layer, which is done in this paper.

In Figure 4, for MNIST, it has only 60% test accuracy. For CIFAR10, it has only 18% test accuracy. This demonstrates my points above. We know closed-form solutions easily for deep neural networks in this paper's setting (as explained above), but this should not work as it is using very strange models so that we can easily have closed-form solutions. Closed-form solutions have no value with 18% test accuracy for simple datasets. It is memorizing the direction of the training data and over-fitting a lot. Linear models work better.

---

> ### Author Response · Authors · 2020-11-16
> **Responses to AnonReviewer3- Part 1**
>
> We would like to thank the reviewer for the feedback and comments. Please see our responses below.
>
> $\textbf{1. [Stationary points vs global optimum]:}$ As the reviewer noted, our results are not directly comparable with existing results characterizing gradient dynamics. Our results characterize the global optimum instead of stationary points or gradient dynamics in terms of closed form formulas. However, this implies that we can obtain the global optimum very efficiently when the assumptions are satisfied. Our experiments suggest that our conclusions also hold for stationary points as well, which requires further theoretical analysis. Please note that although the assumptions may not always be satisfied in practice, the formulas provide an interpretable and intuitive description of the model. We clarify this distinction in the revised version.
>
>
>
> $\textbf{2. [Assumptions on data]:}$ We acknowledge that our assumptions for deep ReLU networks might appear restrictive. However, these assumptions are satisfied in various practical scenarios. First of all, we explore whitened data in Theorem 4.2 (Theorem 4.3 in the revised paper).  Whitening is a common technique in practice, e.g., ZCA whitening in image classification to improve the validation accuracy of the system as empirically shown in [1]. Moreover, we note that the whitening assumption $\bf{X}\bf{X}^T=\bf{I}_n$ necessitates that $n \leq d$. However, the network does not memorize the training set due to the regularization term. In fact our closed form expressions clearly show that this is not the case. Furthermore, the high dimensional setting $n \leq d$ common in few-shot classification and transfer learning problems with limited labels. As an example, it is challenging to obtain reliable labels in problems involving high dimensional data such as in medical imaging and genetics, where $n\leq d$ is typical. There are a multitude of studies applying deep networks to high dimensional data where $n\leq d$ (see references in the main text). More importantly, Stochastic Gradient Descent operates in minibatches rather than the full dataset. Therefore, even when $n>d$, each gradient descent update can only be evaluated on small batches, where the batch size $n_b$ satisfies $n_b \ll d$. Hence, the $n\leq d$ case is implicit during the training phase. Furthermore, recent work has shown that whitening over mini-batches improves the performance of the state-of-the-art architectures, e.g., ResNets, on benchmark datasets such as CIFAR-10, CIFAR-100, and ImageNet [2]. Hence, if we denote the whitened mini-batch of the data samples at a certain iteration as $\bf{X}_b \in \mathbb{R}^{n_b \times d}$, then we have $\bf{X}_b \bf{X}_b^T =\bf{I}$, which is identical to our assumption. Therefore, we believe that our theoretical results for whitened data is also important for practitioners. Finally, we assume that the label matrix $\bf{Y}$ has orthogonal columns. However, in one hot encoded labeling, which is the conventional labeling for classification tasks, the label matrix $\bf{Y} \in \mathbb{R}^{n \times K}$ has orthogonal columns. Hence, classification tasks with one hot encoded labels, e.g., image classification tasks with cross entropy loss, directly satisfy the assumption in Theorem 4.2 (Theorem 4.3 in the revised paper).
>
> $\textbf{3. [Rank one data assumption and non-triviality]:}$ In the revised version, we have clarified our results under rank-one data assumptions separately for two-layer and multi-layer networks. We note that even in this case (two layer networks and rank one data), our results are non-trivial and provides an important generalization of existing results on two layer networks and one-dimensional data (see [3-6]). We respectfully disagree with the reviewer regarding the triviality of rank-one data assumptions since there is now extensive literature on two layer networks with one-dimensional data which is more restrictive than the rank-one data assumption. Note that for two-layer networks, we only assume that the data matrix is an arbitrary rank-one matrix such that $\bf{X}=\bf{c}\bf{a}_0^T$, where $\bf{c}\in \mathbb{R}^n$ and $\bf{a}_0 \in \mathbb{R}^d$. In other words, we do not need the nonnegativity assumption on $\bf{c}$ for two-layer networks. We then use the nonnegativity assumption to extend the analysis to multi-layer networks. However, even in this case, since our analysis is also applicable to deep ReLU networks with bias terms in the last hidden layer (see Proposition 4.1), the activation pattern of ReLU is not the same for all the data samples as claimed by the reviewer. Particularly, in Corollary 4,1 and 4,2, we proved that the optimal output function for two-layer and multi-layer networks are linear spline interpolators for rank-one data and these generalize the two-layer results for one-dimensional data in [3-6] to arbitrary depth.

---

> ### Author Response · Authors · 2020-11-16
> **Responses to AnonReviewer3- Part 2**
>
>
> $\textbf{4. [Practicality of our theory]:}$ In order to show the practicality and significance of our characterization, we have included a new theorem, i.e., Theorem 4.4, proving that all the layer weights can be obtained as a closed-form formula, which is a direct consequence of our characterization in Theorem 4.3. Note that for this case we used the generic form of the problem in Lemma 1.1, which is the regularized training problem. Thus, the optimal solution does not necessarily memorize the labels as in the minimum norm version with equality constraints. More importantly, this result shows that when the data matrix is whitened and the label vector satisfies the conditions in the theorem, which is a common framework especially in image classification tasks, there is no need to train a deep ReLU network in an end-to-end manner. One can directly use the closed-form solutions in Theorem 4.4. to obtain the optimal layer weights.
>
> [1] Coates, Adam, and Andrew Y. Ng. "Learning Feature Representations with K-means.", 2012
>
> [2] Huang, Lei, et al. "Decorrelated batch normalization.", 2018
>
> [3] Savarese, Pedro, et al. "How do infinite width boundednorm networks look in function space?", 2019
>
> [4] Parhi, Rahul and Nowak, Robert."Minimum norm neural networks are splines.", 2019
>
> [5] Ergen, Tolga, and Pilanci, Mert. "Convex geometry of two-layer relu networks: Implicit autoencodingand interpretable models", 2020
>
> [6] Ergen, Tolga, and Pilanci, Mert. "Convex geometry and duality of over-parameterized neural networks", 2020

---

### Official Review · AnonReviewer1 · 2020-10-28
**Closed formed solution for MLPs with ReLU activations**

**Rating:** 6
**Confidence:** 4

**Review:**

This work uses dual formulations of Neural Networks with ReLU activations. It starts explaining the dual formulations with simplest single layer unregularised linear neural networks with a single dimensional output layer. Then gradually extends the models to deep, regularised models with ReLU activations. There is also an assumption on the data to be of rank one or whitened. The experiments are limited and not essential, since they only show that the theory can be confirmed with experiments, albeit they also demonstrate the limitations of simplified models studied here.
This work builds on a recently published work by Ergen and Pilanci, where more limited NNs have been studied, although with a similar dual formulation approach. The main contributions of this paper are the proofs of Theorems 4.1 and 4.2 (given in the appendix). The theorems in section 3 are also novel, but the simplified case of the results in section 4. While very interesting, these results are not applicable in practice, because as the experiments in section 5 show, the models studied here are too simple. However, this is a good step forward towards building a more complete framework to study better NNs. Specifically, it would be interesting to see if a NN with a softmax activation on the output layer can be reformulated with a similar technique.
I did not find the paper easy to read. For example, I haven't found the proof of Lemma 1.1. Several other results have proofs in the appendix, but it is not clear from the main article which proofs are available in the appendix.

---

> ### Author Response · Authors · 2020-11-16
> **Responses to AnonReviewer1**
>
> We would like to thank the reviewer for the feedback and comments. Please see our responses below.
>
> $\textbf{Responses to the comments on the applicability of the results to practical cases:}$
>
> In order to obtain results regrading deep ReLU network, we require certain assumptions that might appear to be restrictive. However, these assumptions are quite common in various practical scenarios. First of all, we assume that the input data is whitened in Theorem 4.2 (Theorem 4.3 in the revised paper).  However, whitening is a common technique in practice, e.g., ZCA whitening in image classification to improve the validation accuracy of the system as empirically shown in [1]. Moreover, we note that the whitening assumption $\bf{X}\bf{X}^T=\bf{I}_n$ necessitates that $n \leq d$. However, this case is common in few-shot classification and transfer learning problems with limited labels. In addition, it is challenging to obtain reliable labels in problems involving high dimensional data such as in medical imaging and genetics, where $n\leq d$ is typical. More importantly, SGD employed in deep learning frameworks, e.g., PyTorch and Tensorflow, operate in minibatches rather than the full dataset. Therefore, even when $n>d$, each gradient descent update can only be evaluated on small batches, where the batch size $n_b$ satisfies $n_b \ll d$. Hence, the $n\leq d$ case implicitly occur during the training phase. Furthermore, recent work has shown that whitening over mini-batches improves the performance of the state-of-the-art architectures, e.g., ResNets, on benchmark datasets such as CIFAR-10, CIFAR-100, and ImageNet [2]. Hence, if we denote the whitened mini-batch of the data samples at a certain iteration as $\bf{X}_b \in \mathbb{R}^{n_b \times d}$, then we have $\bf{X}_b \bf{X}_b^T =\bf{I}$, which is identical to our assumption. Therefore, we believe that our theory (with whitening assumption) is also important for practitioners. Finally, we assume that the label matrix $\bf{Y}$ has orthogonal columns. However, in one hot encoded labeling, which is the conventional labeling for classification tasks, the label matrix $\bf{Y} \in \mathbb{R}^{n \times K}$ has nonoverlapping, therefore orthogonal, columns. Hence, classification tasks with one hot encoded labels, e.g., image classification tasks with cross entropy loss, directly satisfy the assumption in Theorem 4.2 (Theorem 4.3 in the revised paper).
>
> We also provided results with rank-one data assumption. Particularly, in Corollary 4,1 and 4,2, we proved that the optimal output function for two-layer and multi-layer networks are linear spline interpolators for rank-one data, which generalizes the two-layer results for one-dimensional data in [3-6]] to arbitrary depth. We also remark that the analysis of ReLU networks for the one dimensional data considered in these works is non-trivial, which is a special case of our rank-one data assumption.
>
> [1] Coates, Adam, and Andrew Y. Ng. "Learning Feature Representations with K-means.", 2012
>
> [2] Huang, Lei, et al. "Decorrelated batch normalization.", 2018
>
> [3] Savarese, Pedro, et al. "How do infinite width boundednorm networks look in function space?", 2019
>
> [4] Parhi, Rahul and Nowak, Robert."Minimum norm neural networks are splines.", 2019
>
> [5] Ergen, Tolga, and Pilanci, Mert. "Convex geometry of two-layer relu networks: Implicit autoencodingand interpretable models", 2020
>
> [6] Ergen, Tolga, and Pilanci, Mert. "Convex geometry and duality of over-parameterized neural networks", 2020
>
>
> $\textbf{Responses to the comments on Lemma 1.1 and proofs:}$
>
> In the overview section, we presented the problem in its most generic form to simplify the notation. Then, in each section, we first presented the special case of the problem in Lemma 1.1 when $\beta \rightarrow 0$, i.e., the minimum norm version (with equality constraints). We then extend this analysis to the regularized case, i.e., the case with arbitrary $\beta$ as in Lemma 1.1. Hence, in the overview section, we provided Lemma 1.1 as an informal Lemma to motivate our approach and only provided the proofs for each of the cases in the main sections. However, based on the reviewer's comment, we have included a proof for Lemma 1.1 in Section A.3 and added further references for the proof of each result.

---

### Official Review · AnonReviewer4 · 2020-10-28
**Convex duality for finding optimal weights in deep linear and ReLU neural networks.**

**Rating:** 6
**Confidence:** 4

**Review:**

-------------- Overview of the paper -----------------

The paper theoretically studies the structure of optimal weights in deep linear and ReLU neural networks. Using the convex duality formulation, the findings in the paper includes 1) alignment of the weight matrices in deep linear networks; 2) alignment of the weight matrices in deep ReLU networks when the input is rank-one or whitened. Some experiments are provided to verify the theory.

-------------- Contributions and strength -----------------

I think the main contribution of the paper comes from the convex duality formulation of the training problem, which is new. Such an idea allows for extensions of existing works, for example, Savarese et al. (2019), Parhi&Nowak(2019), and Ergen & Pilanci (2020a;b). The corresponding comparison is listed in Table 1.

The paper largely devotes to the theoretical results, while the presentation is clearly organized. I appreciate the warmup section on two-layer linear neural networks, which provides a primary understanding of the more complicated deep networks. I haven't checked every detail of the theory, albeit it appears to be correct and sound in the paper.

-------------- Weakness -----------------

One crucial difficulty in deep learning theory is that deep neural networks are notoriously nonconvex due to their compositional structures. The setup in the paper removes this non convexity (for deep ReLU networks, certain assumption is imposed to enable the duality theory), which undermines the impact of the work. It is not clear to me what are the implications of the theory proposed in the paper to practitioners.

-------------- Further questions and suggestions -----------------

1. The overview section (section 1.1) is a bit deviate from the rest of the section, in that, the training objective is not the minimum norm training problem in the following sections 2 - 4. Lemma 1.1 does not provide understandings of later results.

2. The duality theory requires (X, Y) is feasible, which is a pretty strong condition, given the network is linear. This in turn says the data are noiseless and are obtained from a linear model. An interesting direction to work on is using the dual formulation in Lemma 1.1 to analyze the situation where noise is present.

3. The layout of the paper can be improved:
In Lemma 1.1, the right-hand side of the equation is a bit weird.
In lemma 2.1, the equal sign is used to indicate the two optimization problems are equivalent, which is a bit confusing (reads like the constraint on the left equals the objective on the right).
Figure 2 is not centered compared to Figure 3. The right panel consists of three parallel figures, yet the size is small.

-------------- Minor comments -----------------

Does the training algorithm have an impact on the experimental results, e.g., implicit bias? We observe momentum SGD is used in the MNIST and CIFAR experiment.

———————— update after reading author’s response

I raised my rating to 6 due to the clarification on assumptions in the paper. I am now more convinced of the relevance of the theories in the paper to practice.

---

> ### Author Response · Authors · 2020-11-16
> **Responses to AnonReviewer4- Part 1**
>
> We would like to thank the reviewer for the feedback and comments. Please see our responses below.
>
>
>
> $\textbf{Responses to the comments on the whitening assumption and its practical implications:}$
>
> For the analysis of deep ReLU networks, we assume certain conditions on the training dataset. However, these assumptions are quite common in various practical scenarios. First of all, we assume that the input data is whitened in Theorem 4.2(Theorem 4.3 in the revised paper).  However, whitening is a common technique in practice, e.g., ZCA whitening in image classification to improve the validation accuracy of the system as empirically shown in [1]. Moreover, we note that the whitening assumption $\bf{X}\bf{X}^T=\bf{I}_n$ necessitates that $n \leq d$. However, this case is common in few-shot classification and transfer learning problems with limited labels. In addition, it is challenging to obtain reliable labels in problems involving high dimensional data such as in medical imaging and genetics, where $n\leq d$ is typical. More importantly, SGD employed in deep learning frameworks, e.g., PyTorch and Tensorflow, operate in minibatches rather than the full dataset. Therefore, even when $n>d$, each gradient descent update can only be evaluated on small batches, where the batch size $n_b$ satisfies $n_b \ll d$. Hence, the $n\leq d$ case implicitly occurs during the training phase. Furthermore, recent work has shown that whitening over mini-batches improves the performance of the state-of-the-art architectures, e.g., ResNets, on benchmark datasets such as CIFAR-10, CIFAR-100, and ImageNet [2]. Hence, if we denote the whitened mini-batch of the data samples at a certain iteration as $\bf{X}_b \in \mathbb{R}^{n_b \times d}$, then we have $\bf{X}_b \bf{X}_b^T =\bf{I}$, which is identical to our assumption. Therefore, we believe that our theory (with whitening assumption) is also important for practitioners. Finally, we assume that the label matrix $\bf{Y}$ has orthogonal columns. However, in one hot encoded labeling, which is the conventional labeling for classification tasks, the label matrix $\bf{Y} \in \mathbb{R}^{n \times K}$ has nonoverlapping, therefore orthogonal, columns. Hence, classification tasks with one hot encoded labels, e.g., image classification tasks with cross entropy loss, directly satisfy the assumption in Theorem 4.2 (Theorem 4.3 in the revised paper).
>
>
> $\textbf{Responses to the comments on the rank-one assumption:}$
>
> We believe that our results on rank-one data matrices significantly improves the existing studies on one dimensional datasets. Particularly, in Corollary 4,1 and 4,2, we showed that the optimal output function for two-layer and multi-layer networks are linear spline interpolators for rank-one data, which generalizes the two-layer results for one-dimensional data in [3-6] to arbitrary depth. We also remark that the analysis of ReLU networks for the one dimensional data considered in these works is non-trivial, which is a special case of our rank-one data assumption.
>
>
> $\textbf{Responses to the comments on the practicality of our theory:}$
>
> In order to show the applicability and significance of our characterization, we have included a new theorem, i.e., Theorem 4.4, proving that all the layer weights can be obtained as a closed-form formula, which is a direct consequence of our characterization in Theorem 4.3. Note that for this case we used the generic form of the problem in Lemma 1.1, which also includes cases with noisy observations. More importantly, this result shows that when the data matrix is whitened and the label vector satisfies the conditions in the theorem, which is a common framework especially in image classification, there is no need to train a deep ReLU network in an end-to-end manner. One can directly use the closed-form solutions in Theorem 4.4. to obtain the optimal layer weights.
>
> [1] Coates, Adam, and Andrew Y. Ng. "Learning Feature Representations with K-means.", 2012
>
> [2] Huang, Lei, et al. "Decorrelated batch normalization.", 2018
>
> [3] Savarese, Pedro, et al. "How do infinite width boundednorm networks look in function space?", 2019
>
> [4] Parhi, Rahul and Nowak, Robert."Minimum norm neural networks are splines.", 2019
>
> [5] Ergen, Tolga, and Pilanci, Mert. "Convex geometry of two-layer relu networks: Implicit autoencodingand interpretable models", 2020
>
> [6] Ergen, Tolga, and Pilanci, Mert. "Convex geometry and duality of over-parameterized neural networks", 2020

---

> ### Author Response · Authors · 2020-11-16
> **Responses to AnonReviewer4- Part 2**
>
> $\textbf{ Responses to the further questions: }$
>
> $\textbf{1.}$ In the paper, we focused on the regularized training problem, i.e., $\min \mathcal{L}( f_{\theta,L}(\bf{X}),\bf{y})+\beta  \mathcal{R}(\theta) $, which is the objective in Lemma 1.1. However, in order to simplify the presentation and notation of our main results, we first considered the minimum norm version of this problem, which corresponds to weak regularization, i.e., $\beta\rightarrow 0$. Thus, given the condition $\beta \rightarrow 0$, the problem becomes $\min \mathcal{R}(\theta) $ s.t. $ \mathcal{L}( f_{\theta,L}(\bf{X}),\bf{y})=0$. This shows that the minimum norm training problem is in fact a special case of the generic problem in Lemma 1.1. Since we wanted to provide our results in their most generic form in the overview section, we used the form in Lemma 1.1. Then, for the rest of the paper, we first considered the minimum norm version as a warm-up and then extend the analysis to the regularized version, i.e., the same version with the one in Lemma 1.1.
>
> $\textbf{2.}$ We first note that even for the minimum norm version (with equality constraints) of the problem, feasibility is just a trivial technical condition. In Proposition 2.1, we particularly prove that even in the presence of noise, the optimal solutions to the training problem does not change so that one can assume that the dataset satisfies the feasibility condition without loss of generality. Furthermore, throughout the paper, we also analyzed the regularized versions of the problem, where the feasibility assumption is not required even as a trivial technical condition. Therefore, all our derivations still hold for cases with noisy observations. Based on the reviewer's, comment we have updated the sections with regularized versions to directly include $\bf{y}$ instead of assuming $\bf{X} \bf{w}^* = \bf{y}$. Please see the updated sections 2.1, 2.2, and A.4.
>
> $\textbf{3.}$ We thank the reviewer for pointing out the issues related to the layout, we have revised the corresponding sections as suggested by the reviewer.
>
>
> $\textbf{Responses to the minor comments:}$
>
> For our experiments, we tried both SGD and SGD with momentum, however, we did not observe a significant difference between these algorithms other than the rate of convergence to the optimal solution.

---

### Decision · Program_Chairs · 2021-01-07
**Final Decision**

**Decision:**

Reject

**Comment:**

The paper studies the globally optimal solutions to deep network training problems using convex duality. It derives duals of training problems in which we attempt to fit a dataset while regularizing the network weights with weight decay (L2 regularization). The paper uses strong duality to characterize optimal solutions to several instances of this problem. For fitting deep linear networks, it proves that the globally optimal weights “align” across layers. For fitting ReLU networks, it studies two cases: rank one data and “whitened” data, which satisfy $X^T X = I$. It proves that the optimal weights satisfy certain alignment and orthogonality conditions.

Pros and cons:

[+] The paper uses the machinery of convex duality to characterize alignment of weights in optimal solutions to various neural network training problems. The extension of this approach from shallow networks to deep networks is potentially significant.

[+] The paper is well written and technically precise.

[-] As noted by the reviewers, the assumptions required to analyze deep ReLU networks are somewhat restrictive. In particular, the paper assumes a form of whitening in which the observed data vectors are orthonormal. This is much a much stronger assumption than the whitening usually applied in statistics, in which a linear transformation is applied to ensure that $XX^T = n I$, i.e., the empirical covariance is the identity. While the paper and rebuttal are correct to argue that SGD often uses minibatches of size $n’ \ll d$, the paper’s main claims are about the globally optimal solutions to the overall training problem.

[+/-] Several reviewers raise concerns about the significance of results on the rank-one case for practice. The paper correctly notes that a number of previous works have studied rank one data, and that this paper generalizes those results to deep networks. The paper gives a very clear and explicit recipe for the optimal weights in this restricted setting.  Reviewers are split on the importance of this generalization — in particular, the extent to which results for the rank one case lead to insights that generalize to higher ranks.

[-] Experiments verify the theory, in the sense that the theoretically derived weights are equal or better than those learned by SGD, in terms of the training loss. However, the learned networks do not seem to generalize (right panels of Figure 4), again raising concerns about the realism of the setting.

Reviewers evaluation of the paper is split, with most reviewers appreciating its technical rigor and clean resolution of the rank-1/linear/whitened cases. While the review generated enthusiasm for the paper and its results, there were also significant concerns about the relatively restricted setting and the strength of the paper’s implications for training realistic networks, some of which remain after the authors response.